



# Direct Bayesian model reduction of smaller scale convective activity conditioned on large scale dynamics

Robert Polzin[1], Annette Müller[2], Henning Rust[2], Peter Névir[2], and Péter Koltai[1]

[1]Institute of Mathematics, Free University Berlin, Germany
[2]Institute of Meteorology, Free University Berlin, Germany

**Correspondence:** Robert M. Polzin (robert.polzin@fu-berlin.de)

**Abstract.** We pursue a simplified stochastic representation of smaller scale convective activity conditioned on large scale dynamics in the atmosphere. For identifying a Bayesian model describing the relation of different scales we use a probabilistic approach (Gerber and Horenko, 2017) called *Direct Bayesian Model Reduction* (DBMR). The convective available potential energy (CAPE) is applied as large scale flow variable combined with a subgrid smaller scale time series for the vertical velocity. We found a probabilistic relation of CAPE and vertical up- and downdraft for day and night. The categorization is based on the conservation of total probability. This strategy is part of a development process for parametrizations in models of atmospheric dynamics representing the effective influence of unresolved vertical motion on the large scale flows. The direct probabilistic approach provides a basis for further research of smaller scale convective activity conditioned on other possible large scale drivers.

## 1 Introduction

Complex dynamical processes involving scaling cascades are omnipresent in natural science. Such processes feature different characteristic scales. The smallest and largest scales are far apart and much of the scale range is involved by scale interactions. Dynamics in the atmosphere take place across a large range of time- and length scales, from micro-seconds to months and lengths from $10^{-5}$ to $10^6$ m. Due to the geostrophic and hydrostatic equilibrium there is a scale separation induced by thermal stratification, gravity and rotation for scales above several kilometers (Klein, 2010). Thunderstorms last a few tens of minutes for example, whereas hurricanes may last for days. Medium-range forecasts are made up to 10 days in advance. Predictions of convection further in advance cannot be deterministic and are highly uncertain because errors of the initial space of the smaller scales are growing.

A new perspective for improving General circulation models (GCMs) came from parameterizations. An example are parameterizations that represent the small scale effects of convection on the large-scale dynamics (Berner et al., 2017; Franzke et al., 2015). Nowadays, many data-driven approaches are dwelling on stochastic parametrization methodologies involving the convective available potential energy (CAPE) as large scale driver for convection, e.g. in (Khouider et al., 2010; Dorrestijn et al., 2013a, b). Their approaches need high computing capacities, but the costs to process large quantities of data can become a limiting factor. The statistical analysis of atmospheric dynamics simulations requires dimensionality reduction techniques which





yield applicable reduced models. One way is the Empirical orthogonal function (EOF) analysis which is a tool for data compression and dimensionality reduction used in meteorology. Since its introduction by Lorenz (1956), EOF analysis—known as principal component analysis (PCA) or proper orthogonal decomposition (POD)—has become an important statistical tool in atmosphere science. For example in Horenko et al. (2008) different sets of EOFs are used for a reduced representation of meteorological data. The applicability of many approaches is based on the identification of reduced models defined on a small set of

latent states. These methods derive aggregations of original variables based on a reduced approximation of the system in terms of relation matrices. Examples are covariance matrices (Schölkopf et al., 1997; Jolliffe, 2003), partial autocorrelation matrices of autoregressive processes (Schmid, 2010), Gaussian distance kernel matrices (Donoho and Grimes, 2003; Coifman et al., 2005), Laplacian matrices as in the case of spectral clustering methods for graphs (Von Luxburg, 2007), adjacency matrices in community identification methods for networks (Zhao et al., 2012). A recent algorithmic framework called Direct Bayesian

Model Reduction (DBMR) (Gerber and Horenko, 2017; Gerber et al., 2018) provides a computationally scalable probability-preserving identification of reduced models and latent states directly from the data. The method constructs a directly low-rank transition matrix, reducing numerical effort and estimation error due to finite data. The latter approach does not require a distributional assumption but works instead with a discretized state vector. Our aim is the development of a model combining the deterministic large scale atmospheric flow with a conceptual stochastic description of small scale convection. Towards this

goal, we develop a conceptual categorical description for smaller scale vertical velocity, which is linked to a large scale flow variable. The probabilistic description is proposed using DBMR. Relation between the probability for large scale and smaller scales can be formulated categorically via a conditional probabilities and the conservation of the total probability. Various energetic variable are applicable on large scale. Other potential large scale variables driving the smaller scale stochastics besides CAPE are the Dynamic State Index (DSI) (Müller et al., 2020; Müller and Névir, 2019), available moisture, or vertical wind

shear. The DSI is a scalar diagnostic field that quantifies local deviations from a steady and adiabatic wind solution and thus indicates non-stationarity as well as diabaticity.

The paper is structured as follows: In Sect. 2 the mathematical methodology of DBMR is presented. Afterwards, in Sect. 3 the set-up for a reduced model in the atmosphere is described. In Sect. 4 the results are discussed related to atmospheric dynamics. Finally, in the conclusion the results and future work towards the direct Bayesian model reduction of smaller scale

convective activity conditioned on large scale dynamics are formulated.

## 2   Mathematical methodology

Our aim is to study and understand a stochastic relation between two variables $X$ and $Y$ that can take values from two finite sets. These categorical random variables will later on encode quantitative information of the atmosphere on different spatial scales. We will review a novel computational framework for the estimation of a reduced (low-rank) Bayesian model from

data. This method is called Direct Bayesian Model Reduction (DBMR). Direct refers to a directly low-rank estimation which is useful for the identification of reduced models, yielding thereby an advantageous estimation error, especially if data is not abundant (Gerber and Horenko, 2017).





## 2.1 Stochastic model

We are interested in modeling the probabilistic relationship of two potentially random quantities, $X$ and $Y$. For us, it will be
only relevant that $Y$ is a random function of $X$—randomness of $X$ itself is irrelevant. Since the observations typically arise as
time series, we consider $X$ and $Y$ as processes $X(t)$ and $Y(t)$ with time $t$, however $t$ can denote any parameter ordering the
realizations of the process. We will consider the case where $X$ and $Y$ can only attain a finite number of values, such that we
call the processes discrete-state or categorical. Say, $Y(t)$ is taking one of the possible values from $m$ categories $\{y_1, y_2, ..., y_m\}$
and $X(t)$ from the $n$ categories $\{x_1, x_2, ..., x_n\}$. The central quantity of interest describing the relationship of $X$ and $Y$ is the
$m \times n$ matrix of conditional probabilities, also called transition matrix,

$$\Lambda = \begin{pmatrix} \mathbb{P}[Y = y_1 \,|\, X = x_1] & \cdots & \mathbb{P}[Y = y_1 \,|\, X = x_n] \\ \vdots & \ddots & \vdots \\ \mathbb{P}[Y = y_m \,|\, X = x_1] & \cdots & \mathbb{P}[Y = y_m \,|\, X = x_n] \end{pmatrix}. \tag{1}$$

Note that $\Lambda$ is a column-stochastic matrix. In practical studies, when the $\Lambda_{ij}$ are estimated from the available observations of
$X$ and $Y$ one needs to guarantee that the data is acceptably randomised (Holland, 1986). We will assume that

$$\mathrm{Law}\left[Y(t) \,|\, X(1), X(2), \ldots\right] = \mathrm{Law}\left[Y(t) \,|\, X(t)\right], \tag{2}$$

i.e., given the input $X(t)$, the distribution of the output $Y(t)$ is independent of the other inputs.

## 2.2 Bayesian approach

Typically, the transition matrix $\Lambda$ is not directly available and can only be estimated from observed data. Let $S$ be the number
of observation pairs for the categorical processes $X$ and $Y$, such that the following observational data is available:

$$\boldsymbol{XY} = \{X(1), X(2), ..., X(S), Y(1), Y(2), ..., Y(S)\}, \tag{3}$$

where $X(t) \in \{x_1, \ldots, x_n\}$ and $Y(t) \in \{y_1, \ldots, y_m\}$, as above. Given $\boldsymbol{XY}$, it is reasonable to search for the $\Lambda$ for which the to-
tal probability of obtaining the particular sequences of observations (3) is maximized. By the independence assumption (2), the
*likelihood* of a matrix $\Lambda$ of conditional probabilities—i.e., the probability of observing the data if the conditional probabilities
were given by $\Lambda$—is given by

$$\mathbb{P}[\boldsymbol{XY} \,|\, \Lambda] \propto \prod_{i=1}^{m} \prod_{j=1}^{n} \underbrace{\mathbb{P}[Y = y_i \,|\, X = x_j]}_{=\Lambda_{ij}}^{N_{ij}}, \tag{4}$$

where $N_{ij}$ is the total number of instances in the data when $\big(X(t), Y(t)\big) = (x_j, y_i)$. The optimum can be more easily com-
puted if one considers the *log-likelihood* $\log(\mathbb{P}[\boldsymbol{XY} \,|\, \Lambda]) = \sum_{i=1}^{m} \sum_{j=1}^{n} N_{ij} \log \Lambda_{ij}$, with which we arrive at the maximum
likelihood problem

$$\Lambda^* = \arg\max_{\Lambda} \left\{ \sum_{i=1}^{m} \sum_{j=1}^{n} N_{ij} \log \Lambda_{ij} \right\}, \text{ such that } \Lambda_{ij} \geq 0, \sum_{i=1}^{m} \Lambda_{ij} = 1. \tag{5}$$



The optimal solution of that constrained optimisation problem can be determined analytically (Gerber and Horenko, 2017), resulting in the empirical frequency estimator:

$$\Lambda_{ij}^* = \frac{N_{ij}}{\sum_{j}^{n} N_{ij}}.$$  (6)

Since we merely have a finite amount of observation at hand, it is essential to be aware of the uncertainty of the statistical estimate (6). While we refer the reader interested in exact bounds to (Gerber and Horenko, 2017, Supplement, Eq. (14)), an intuition can be gained as follows. To estimate each $\Lambda_{ij}$ to a sufficient (statistical) accuracy, the transitions $N_{ij}$ should be, on average, numerous. As there are $nm$ parameters in $\Lambda$ to estimate, this asks for the sample size $S$ to be reasonably large as compared to $nm$. In practice, this can be problematic if $n$ and $m$ are large. Thus, next we will discuss a modification of the above method that can mitigate this problem.

## 2.3 Direct estimation of low-order models

In numerous situations the apparent complexity of our observations is an artefact of our measurement procedure, and there are low-dimensional features that govern the process at hand. Thus, even if we would be able to find a full matrix $\Lambda$ of conditional probabilities, the ultimate goal would be to reduce this through such low-dimensional features.

The following approach, proposed by Gerber and Horenko (Gerber and Horenko, 2017), achieves both estimation and reduction in one step. We *assume* that the output depends on the input through a *latent variable* $Z$, which can merely take a small number $K \ll \min\{n,m\}$ of different states $\{z_1, \ldots, z_K\}$. In terms of probabilistic influences, we assume the structure

$$X \xrightarrow{\Gamma} Z \xrightarrow{\lambda} Y,$$  (7)

where $\lambda, \Gamma$ are matrices of conditional probabilities,

$$\Gamma_{kj} = \mathbb{P}\big[Z = z_k \,|\, X = x_j\big], \qquad \lambda_{ik} = \mathbb{P}\big[Y = y_i \,|\, Z = z_k\big].$$  (8)

We also assume conditional independence of $Y$ on $X$ given $Z$, that is, the input-output conditional probability matrix $\Lambda$ satisfies $\Lambda = \lambda\Gamma$. Note that we can interpret $\Gamma_{kj}$ as an *affiliation* of input category $x_j$ to the latent state $z_k$, see Fig. 1.

The task is now to determine the pair of column-stochastic matrices $(\lambda, \Gamma)$ from the observation data $\boldsymbol{XY}$, as given in (3). Again, we wish to solve the problem with a maximum-likelihood approach, which would require solving (5) with replacing $\Lambda_{ij}$ by $(\lambda\Gamma)_{ij}$ and the constraints by requiring $\lambda$ and $\Gamma$ being stochastic matrices. This, however, is a computationally hard optimization problem, which Gerber and Horenko in (Gerber and Horenko, 2017) relax to

$$(\lambda^*, \Gamma^*) = \arg\max_{\lambda, \Gamma} \sum_{i=1}^{m} \sum_{j=1}^{n} \sum_{k=1}^{K} N_{ij} \Gamma_{kj} \log\big\{\lambda_{ik}\big\}$$  (9)

subject to

$$\lambda_{ik} \geq 0, \sum_{i=1}^{m} \lambda_{ik} = 1, \qquad \Gamma_{kj} \geq 0, \sum_{k=1}^{K} \Gamma_{kj} = 1.$$  (10)





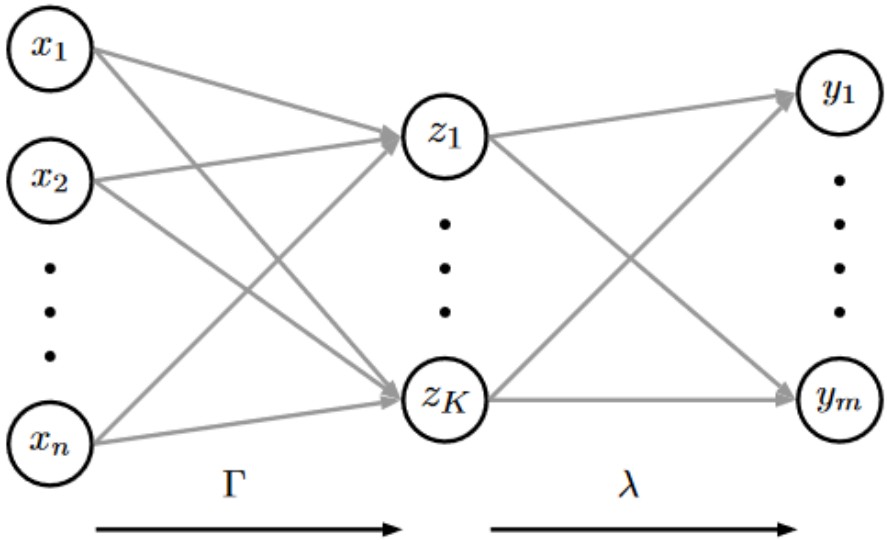

**Figure 1.** Introduction of intermediate latent states in DBMR for efficient and scalable estimation of $\Lambda$

While (9) produces suboptimal estimates, its advantage comes from the fact that it is concave in both variables $\lambda$ and $\Gamma$, respectively, allowing for a very simple alternating maximization as optimization procedure (Gerber and Horenko, 2017, DBMR algorithm). The resulting algorithm is DBMR. Moreover, the method yields $\Gamma_{kj}^* \in \{0, 1\}$, i.e., the original input categories are assigned to the reduced system's (latent) categories in a deterministic fashion (no "fuzzyness" in the affiliations). Of course, the number $K$ of latent states is not known in advance, and has to be chosen judiciously by compromising between "expressiveness" (the likelihood of the model, i.e., the optimal value in (9)) and "effort" (the number of total parameters to be estimated and their statistical error). This can be done comparing multiple DBMR runs with different $K$.

The obtained models are also less subject to overfitting issues and are more advantageous in terms of the model quality measures (Gerber and Horenko, 2017; Gerber et al., 2018). This manifests in the variance of the estimated parameter $\lambda_{ik}^*$, which shows a $K/n$-times smaller uncertainty than $\Lambda_{ij}$, cf. (Gerber and Horenko, 2017, Theorem and eqn. [7]). Again, intuitively this advantage of DBMR over the full model (6) can be seen by noting that from the same amount of data DBMR only needs to estimate $k(n + m)$ parameters, while the full model $nm$ parameters.

Let us emphasize that additionally to all the computational advantages of DBMR that allow it to work with large data sets, its conceptual strength is that it combines model estimation and model reduction in one step. The latent states often have a physical meaning—a property that we shall focus on in our application.



## 3 Data pre- and postprocessing

To apply DBMR, categorical processes for the in- and output have to be defined. First, we discuss the choice of meteorological variables and scales in view of the categorical processes. As input we use a variable related to large scale atmospheric flow:
*Convective Available Potential Energy* (CAPE), a measure for the energy an air parcel would gain if lifted to a specific height in the atmosphere.

### 3.1 Meteorological data

CAPE can be seen as a measure for atmospheric stability, first suggested by (Weisman and Klemp, 1982). It is defined by

$$CAPE = g \int\limits_{z_{\mathrm{LFC}}}^{Z_{\mathrm{ET}}} \frac{\theta_e - \theta}{\theta} \, dz \,, \tag{11}$$

where $\theta_e$ is the pseudopotential temperature of the ascending air parcel, $\theta$ is the potential temperature of the surrounding air, and $z_{\mathrm{LFC}}$ is the so-called *Level of Free Convection* (LFC). The LFC is the height at which the rising air parcel becomes significantly warmer than its environment; $Z_{\mathrm{ET}}$ denotes the height, where the rising air parcel has the same temperature as its environment (ET stands for equal temperature). Thus, regarding its definition (11), CAPE becomes large if the temperature difference between the rising air and the environmental air is large, see (Bott, 2016, p. 431 ff). As an integral, CAPE is a global
variable that we consider as representative variable on the larger scale. To capture convective activity, characterized by strong up- and downdrafts, on the smaller scale, we regard the vertical velocity. Parcel theory (Dutton, 1976) predicts

$$\mathrm{CAPE} \sim \frac{v_{max}^2}{2}, \tag{12}$$

where $v_{max}$ is the maximum vertical motion in the dimension m/s expected from the release of CAPE in the dimension J/kg. The relation in Eq. (12) is a kinetic description of a potential which does not have to be released to vertical updraft.
(Moncrieff and Miller, 1976) were the first to use the term CAPE. The USAF Air Weather Service (which changed its name to the Air Force Weather Agency in 1997) simply called it positive area (AWS 1961). (Fritsch and Chappell, 1980) called it potential buoyant energy (PBE), while variations of this include +BE and net positive buoyant energy. Despite the abundance of names, it now appears that CAPE is the de facto standard terminology. In (Kirkpatrick et al., 2009) over 200 convective storm simulations are analyzed to examine the variability in storm vertical velocity and updraft area characteristics as a function of
basic environmental parameter CAPE.

To analyze the relation of large and small scale parameters, the COSMO-REA6 reanalysis data set is used (Bollmeyer et al., 2015). This reanalysis is based on the non-hydrostatic numerical weather prediction model COSMO (*COnsortium for Small scale MOdelling*) by the German Meteorological Service (Deutscher Wetterdienst, DWD) using a continuous nudging scheme. It has a horizontal resolution of 6 km and 40 vertical layers (Bollmeyer et al., 2015). Since we focus on smaller scale
convective events conditioned on large scale dynamics in the atmosphere, we consider the summer months July and August in the years 1995 to 2015. For our analysis we use 12h means. The sample size of the reanalysis data set used in Sect. 2

sums up to $S = 1302$ ($2 \times 31 \times 21$). We choose a REA6 subdomain that covers Germany. This subdomain is bounded by the $45.2°N - 54.7°N$, $5.8°E - 15.3°E$ and shown in Fig. 2. The Northwest coordinate is $(5.8°E; 54.7°N)$ and the Southeast coordinate is $(15.3°E; 45.2°N)$. As vertical layer the 600 hPa surface is considered, because here the latent heat release takes

place and the vertical velocity reaches its maximum, as shown in (Müller et al., 2020).

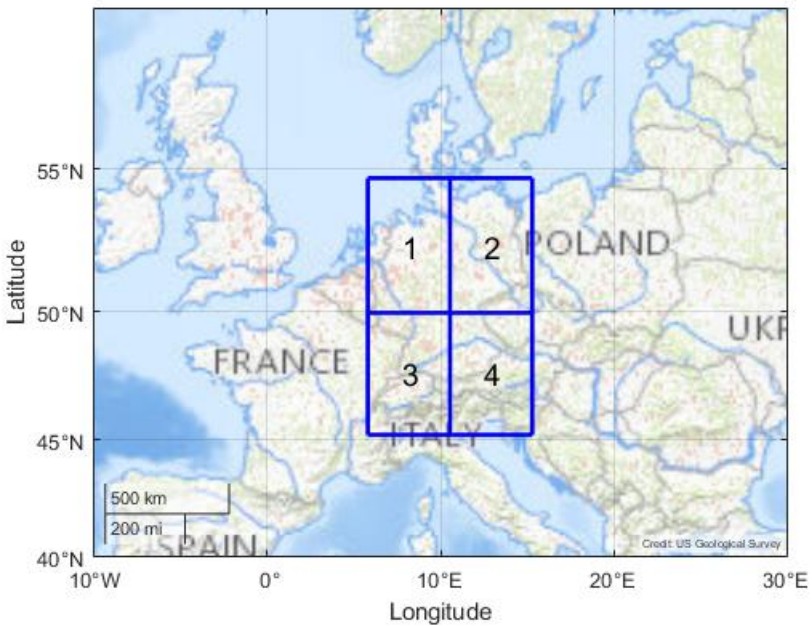

**Figure 2.** REA6 domain that covers Germany consisting of subdomains 1 to 4; Subdomain 1 is applied on the large scale for DBMR and is of approximately $500\,\mathrm{km} \times 500\,\mathrm{km}$. Image credit of the map: US Geological Survey (USGS).

### 3.1.1   Filtering CAPE and vertical velocity

The domain that covers Germany in Fig. 2 is divided into four $500\,\mathrm{km} \times 500\,\mathrm{km}$ quadrants, where the spatial arithmetic mean of each of the quadrant is considered such that we obtain one CAPE value for each quadrant. We separate and filter the data of CAPE and the vertical velocity in further subdomains in order to define the categorical in- and output. The corresponding sizes

of the subdomains are summarized in Tab. 1. For the analysis with DBMR, the northwest quadrant 1 over Holland in Fig. 2 is used. There is no influence of the Alps on smaller scale convective activity.

### 3.2   Categorical input and output

According to the meteorological data in Sect. 3.1, we will set up applicable categories for in- and output. CAPE plays the role of an *input* variable $X$ in Sect. 2, describing the potential for convection. We use the average of the $500\,\mathrm{km} \times 500\,\mathrm{km}$

quadrants, considering CAPE as the large scale atmospheric driver. With energy units, CAPE has a non-negative range of





| # Subdomains m | Edge length |
|---|---|
| $1^2 = 1$ | 1000 km |
| $2^2 = 4$ | 500 km |
| $4^2 = 16$ | 250 km |
| $6^2 = 36$ | 167 km |
| $8^2 = 64$ | 125 km |
| $10^2 = 100$ | 100 km |
| $12^2 = 144$ | 83 km |
| $16^2 = 256$ | 63 km |
| $32^2 = 1024$ | 31 km |
| $64^2 = 4096$ | 15 km |

**Table 1.** Number of subdomains $m$ and edge length for box discretization of the atmosphere from large synoptic scale (1000 km) across intermediate scales up to meso-gamma scale with convective activity $(2 - 20 \, \text{km})$

values. The model's *output* variable $Y$ is vertical velocity obtained on a smaller scale. Here, $Y$ can take positive and negative values for updrafts and downdrafts, respectively. We average over $250 \, \text{km} \times 250 \, \text{km}$ to $15 \, \text{km} \times 15 \, \text{km}$ according to Tab. 1.

### 3.2.1 Categorical input

We will consider two different ways to define the input categories. First, linearly spaced categories for the range of CAPE

are applied. With this type of classification, extreme weather events tend to be in a separate category. These are not Gaussian distributed. Alternatively, the categorization by evenly spaced quantiles is presented. We consider the range of values for CAPE $(X)$ and generate $n$ categories by $\{x \in X_i \mid b_{i-1} \leq x < b_i\}$. For the category boundaries $b_i$, we consider the following equally spaced options

- on the linear CAPE axis with $b_i = \min \text{CAPE} + i \frac{\max(Y) - \min(Y)}{n}$, $i = 0, \ldots, n$,

- in probability using empirical $1/n$-quantiles as category boundaries.

While the first is easily interpreted on the CAPE-axis, the second categorization has the advantage of (almost) equally populated categories. The resulting $n$ categories are denoted by integers $1, \ldots, n$, we choose $n = 10$. In Sect. 3.1 we set up the meteorological data with a size of the observational data $S = 1302$. That means we have about 130 data points in each CAPE category.

### 3.2.2 Categorical output

We map vertical velocities $\omega_i$ at grid box $i$ on a variable $Y_i \in \{1, 2, 3\}$ as



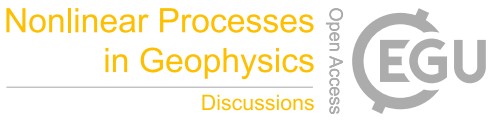

- **updraft** for $Y_i = 1$, if $\omega_i \geq a_2$,

- **no draft** for $Y_i = 2$, if $a_1 \leq \omega_i < a_2$,

- **downdraft** for $Y_i = 3$, if $\omega_i < a_1$,

$(a_1, a_2) \in \mathbb{R}^2$ define a potentially asymmetric interval around zero vertical velocity which we consider as neutral, with $a_1 < 0$ and $a_2 > 0$. The choice of $(a_1, a_2)$ depends on the scale of the box where $Y$ is averaged over. In Sect. 4.1 the choice of the interval for our analysis is described. Once this discretization is made, the final output categories needed can be set up. Let $Y_i(t)$ be the discretized vertical velocities at time $t$ with $1 \leq i \leq m$ numbering the grid boxes on the corresponding scale, see Tab. **??**. We define the following categorical process

$$\hat{Y}(t) = (\#\{Y_i(t) = 1\}, \#\{Y_i(t) = 2\}, \#\{Y_i(t) = 3\}) \in \mathbb{N}^3, \tag{13}$$

with $\#\{Y_i = k\}$ being the number of grid boxes with vertical velocity mapped onto $k \in \{1, 2, 3\}$. There are exactly $(m+1)^2$ ways to decompose $m$ into the (ordered) sum of 3 nonnegative numbers, thus the number of actually occurring categories $n_{\hat{Y}} \leq (m+1)^2$. In our probability-preserving algorithm the number of the occurring categories in the data are counted for the categorical observational input and output. The probability of a category is estimated by its occurrence frequency with respect

to the total number of data points. We try to conclude down- and updraft behavior from the $\hat{Y}(t)$, i.e. the distribution of up- and downdrafts. Note that we have in this experiment no information on the (spatial) structure, as the category in (13) is a triple of numbers for counts of down-, updraft and low vertical velocity.

### 3.3 Maximum likelihood estimation

The model reduction is a consequence of using the affiliation matrix $\Gamma$, which assigns the $n$ large scale categories to $K < n$

latent states. In the frame of DBMR we optimize a relaxed log-likelihood, cf. (9). We ran DBMR 100 times (with random initializations) for every fixed number $K$ of latent state. For the respective latent state, the run with the maximum log-likelihood is presented. We also evaluate the exact log-likelihood, as in (5). Fig. 3 shows the exact in blue and the relaxed log-likelihood in red, both for the reduced problem, i.e., the one with latent states. The only parameter in the algorithmic procedure introduced above is the reduced process dimension $K$ for the number of collective causality boxes. It can be chosen by comparing results

for different $K$ and selecting the best reduced model according to one of the standard model selection criteria (Cross-validation with a performance criterion, Akaike Information Criterion (AIC), Bayesian Information Criterion (BIC) or L curve approach). For an attempt for model selection, the largest increase in log-likelihood can be found by increasing $K = 2$ to $K = 3$, for $K = 6$, the maximum value has been reached. Note that as $n = 10$, choosing $K = 10$ presents no model reduction.

## 4 Reduced Bayesian model for atmospheric dynamics

### 4.1 Results

In the following, the pre- and postprocessing on DBMR with respect to the categorical in- and output will be discussed.


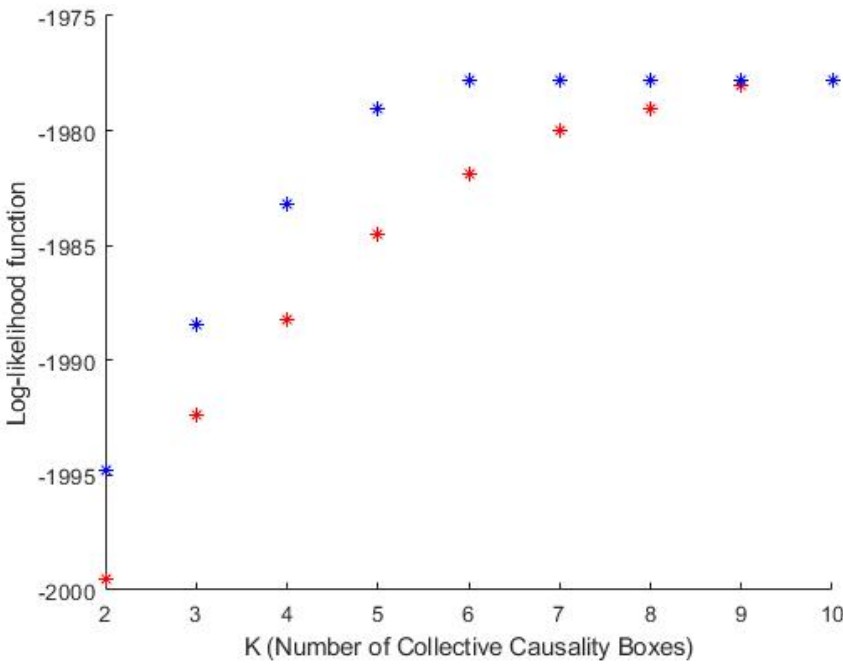

**Figure 3.** Exact log-likelihood value as in (5) (blue) and relaxed log-likelihood value as in (9) (red) of the reduced Bayesian model estimated by DBMR with $K$ latent states (also referred to as collective causality boxes).

### 4.1.1 Interval for vertical draft

All-day mean data serve as basis for determining the interval for vertical draft. The subclassification by the interval is symmetric with $a_1 = -0.0048$m/s and $a_2 = 0.0048$m/s. These values are chosen from a histogram of mean vertical velocities. In Fig. 4,
the histogram of mean vertical velocities for a resolution of 125km with the interval is shown.

Afterwards, the data for day and night are split up and will be fed to DBMR respectively.

### 4.1.2 Affiliation to latent states

In Fig. 5 the summary statistics with boxplots and images of the affiliation of input categories to the latent states are visualized for 2 latent states. This is done for day and night, respectively. For every latent state a boxplot is provided. The latent states are
interpreted as reduced units of the large-scale atmospheric state with respect to their probabilistic impact on vertical motion. On each box, the central mark indicates the median, and the bottom and top edges of the box indicate the 25th and 75th percentiles of CAPE concerning the affiliated $n$ categories ($n = 10$ divided by quantiles). In Fig. 5 one sees that the input categorization is similar in terms of value for day and night. The first latent state includes 5 (for day) and 4 (at night) CAPE categories with high values. This represents high CAPE values and is therefore referred to as "High". Five (for day day) and 6 (at night) categories



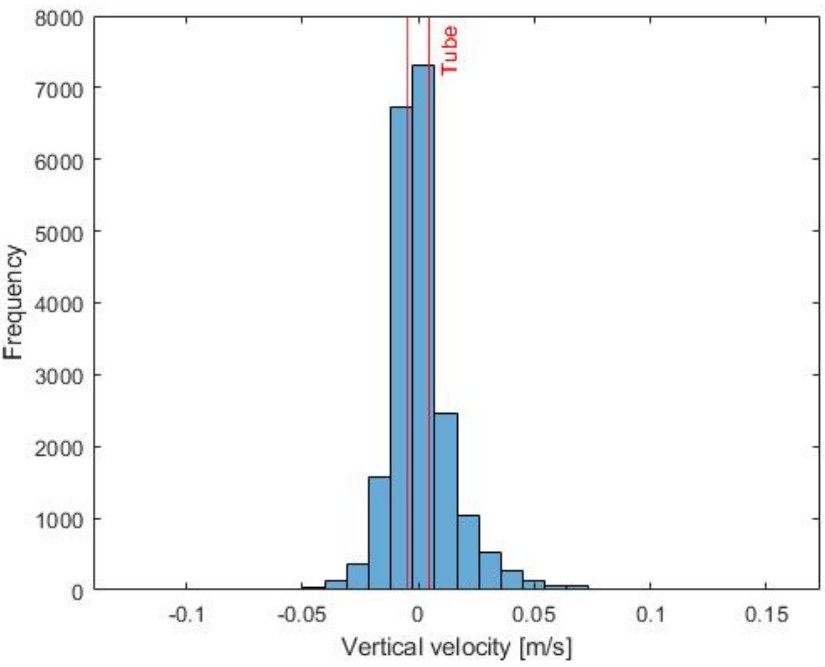

**Figure 4.** Histogram of mean vertical velocities for day and night for resolution of 125 km and hourly averaged values; red vertical lines represent a tube for vertical draft; The summer months July and August in the years 1995 to 2015. For our analysis we use 12h means. The sample size of the reanalysis data set sums up to $S = 1302$ $(2 \times 31 \times 21)$.

are affiliated to the second latent state, which is denoted with "Low". During daytime the categories reach values up to 386 J/kg, whereas at night the values have a range of 343 J/kg due to less convective activity. The affiliations to the latent states have no gaps for day and night. That means the latent states are separate from each other, see boxplots in Fig. 5. The difference between the scales is small (375 km) with 500 km step size on large scale and 125 km step size on the smaller scale. The scale jump is of factor 4 on the basis of the small scale. The results for three latent states are shown in Appendix A. The third latent

state represents mean CAPE categories. At night, five categories are even affiliated to the latent state "Mean", see Fig. A1. In the following, the visualization of the output is discussed, in the appendix further results on different CAPE categories and affiliations are shown.

### 4.1.3 Distributions conditioned on latent states

We discuss probability distributions conditioned on the resulting latent states introduced in Sect. 2.3 in two ways:

– Law$[X \mid Z]$ gives the distribution of CAPE $X$ within a latent state $Z$,

     – Law$[\#1, \#3 \mid Z]$ gives the joint probability distribution of number of grid points with positive and negative vertical velocity. For updraft, $\#1$ denotes $\#\{Y_i = 1\}$ and for downdraft, $\#3$ denotes $\#\{Y_i = 3\}$.


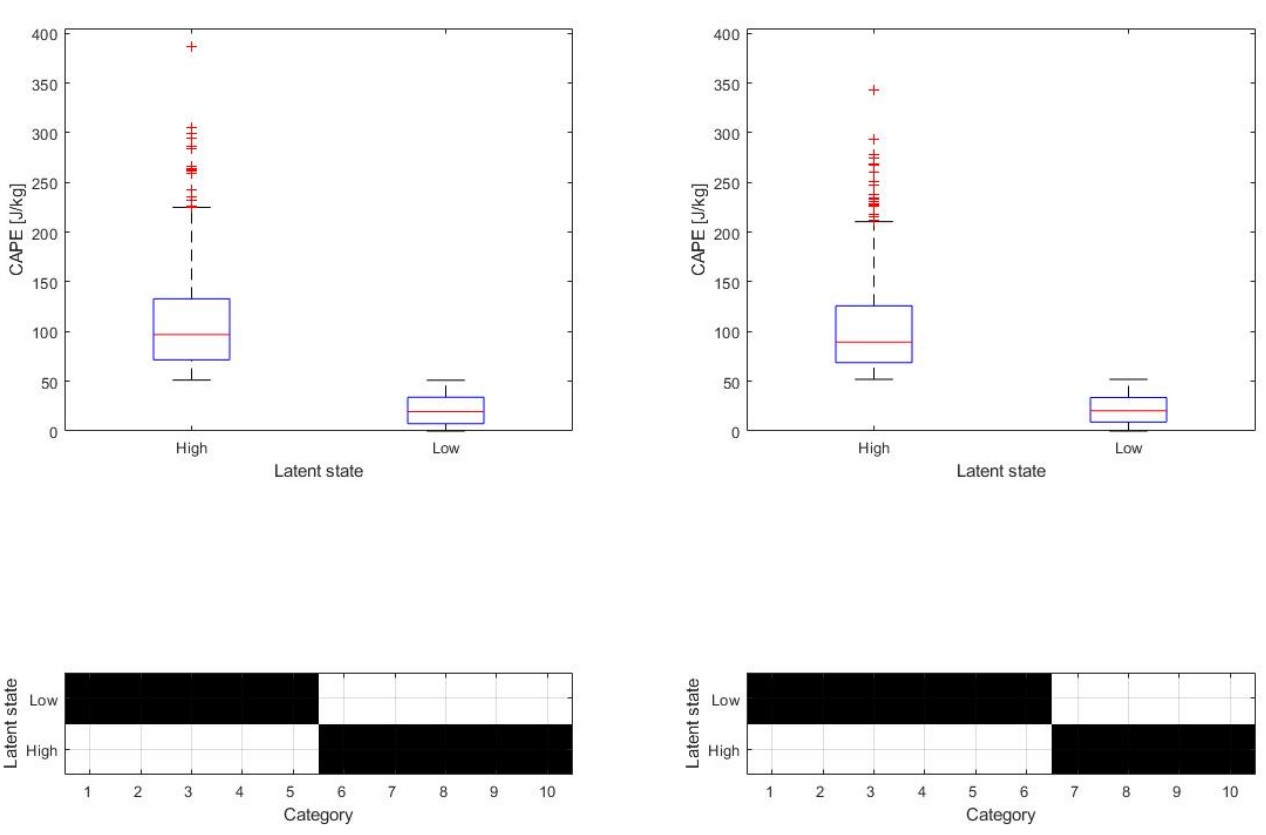

**Figure 5.** Top: Boxplot of CAPE categories by 2 latent states for daily mean (left: day and right: night); Bottom: Affiliation of CAPE categories to the latent states; Data on large scale: $500\,\mathrm{km} \times 500\,\mathrm{km}$ subdomains for northwest of Germany, hourly averaged CAPE; On mesoscale: $125\,\mathrm{km} \times 125\,\mathrm{km}$ subdomains for vertical velocity. Time series: 21 years for July and August, $S = 1302$ (Length of time series)





The missing number of neutral grid points $\#\{Y_i = 2\}$ follows from $\#2 = m - \#1 - \#3$ with $m$ denoting the total number of grid points. In order to visualize the probabilities of the small scale conditioned on the latent states, the entries of the $\hat{\lambda}$

matrix in (9) will be displayed dependent on the number of down- and updrafts. In Fig. 6, $K$ bivariate histograms are shown for day and night respectively. Here the conditional probabilities of matrix $\lambda$ are displayed for every latent state dependent on the number of up- and downdrafts (#1 and #3). Since the number of smaller-scale boxes is $n_Y$, only the lower triangle below the diagonal corresponds to categories. Categories not populated by data are not shown (white). We noticed that in case the interval for vertical draft in Sect. 4.1 are increased, fewer data points are in the subclassifications for the up- and downdrafts

(i.e. smaller numbers #1 and #3 change lower triangular probability matrices of Fig. 6). A comparison of different sizes of intervals for vertical draft is not shown here. Increasing the interval makes less up-/downdrafts, thus moving probability mass away from the diagonal, where large fractions of up-/downdrafts are sitting. In Fig. 6 the results are shown for a $4 \times 4$ grid, that means we have 16 boxes with vertical velocities. In the histograms the numbers of up- and downdraft range from 0 to 16. Variable $Z_1$ represents the latent state "High", latent state $Z_2$ the state "Low", as in Fig. 5. The latent states are stochastically

disaggregated in probabilities which describe the chance of number of up- and downdrafts conditioned on the latent states "High" and "Low". In the top left panel ($Z_1$, day) of Fig. 6 probability adds up for numbers of up- or downdrafts higher than 10 to 81%. In the top right panel, probability accumulates at small numbers of boxes with downdraft. For Law $[\#1, \#3 \mid Z_2]$ in the bottom left panel for the day, high conditional probabilities $\mathbb{P}[\#1, \#3 \mid Z_2]$ concentrate in categories with many boxes with downdraft. Here the probability of numbers of downdrafts of 6 to 16 is 68%. At night in the latent state $Z_2$, we observe

that a low number of updraft boxes is likely, while the overall up- and downdraft activity seems to be the least probable here (probability concentrating around $(\#1, \#3) \approx (0,0)$). In the bottom left panel ($Z_2$, night) the probability is accumulated to 82% for the number of updrafts between 0 and 4.

**Figure 6.** Probabilities of numbers of updrafts (#1) and downdrafts (#3) conditioned on latent states $P[\#1, \#3|Z]$; Day (left) and night (right) and Latent state "High" (top) and latent state "Low" (bottom); $500\,\mathrm{km} \times 500\,\mathrm{km}$ domain for CAPE and $125\,\mathrm{km} \times 125\,\mathrm{km}$ subdomains for vertical velocity.





### 4.1.4   Output on smaller scale

Note that the number of possible output categories $\hat{Y}$ scales quadratically with the number $m$ of grid points considered on the
smaller scale. Moving towards the convective scale, $m$ increases, and so does the number of possible output categories, yet the
number of data points (1302) stays the same. To avoid the resulting increase of estimation error, we further reduce the number
of output categories by dividing the respective numbers for up- and downdraft into 3 sections, which leaves 6 categories. We
use $15\,\text{km} \times 15\,\text{km}$ subdomains on convective scale for the output of DBMR. The large scale remains unchanged compared to
the previous example. In Fig. 7 the distribution of CAPE in terms of latent states based on kernel density estimation (KDE)
is shown. KDE is a non-parametric way to estimate the probability density function of a random variable. At night, more
categories are assigned to the latent state "Low", the first latent state has a larer mean and median than during daytime.

In Fig. 8 the conditional probabilities are shown for 1024 boxes of vertical velocities. In the histograms the 3 sections of
numbers of up- and downdraft range from 0 to 1024. The 3 categories are divided by the following numbers: 0 to 341, 342 to
683 and 684 to 1024 up- and downdrafts. Variable $Z_1$ represents again the latent state "High", and $Z_2$ the latent state "Low";
cf. Fig. 8. The first latent state is represented in the first row. During daytime down- or updraft is likely, and during nighttime
it is most likely to have less downdraft than updraft. The smaller scale analysis gives consistent results with the analysis where
the output is on mesoscale in Fig. 6. There are higher probabilities during daytime for medium to high numbers of up- or
downdraft. At night due to less vertical draft, low to medium numbers of up- or downdraft are higher. For the second latent
state "Low", the distributions concentrate on higher and lower numbers of downdrafts and small numbers of updraft.

### 4.2   Higher number $K$ of latent states

The results for three latent states are considered in Appendix A. Figs. A1 and A2 show results using CAPE as input with a
resolution of $500\,\text{km} \times 500\,\text{km}$ on large scale and a grid of $125\,\text{km} \times 125\,\text{km}$ for the output. The scale difference is again of
factor 4 according to the first example in Sect. 4.1 where in- and output are on the synoptic scale. Affiliations without gaps
lead to a separation of the latent states. "No gaps" means that affiliations are interrelated and not interrupted in the middle
plots of Fig. A1 and Fig. A3. The affiliations have no gaps for day and night. We have again more variance of the conditional
probabilities during daytime. At night there is less variance of the conditional probabilities with a concentration at low numbers
of downdraft or updraft boxes. A hierarchy of three different probability configurations arises for up-, down- and no draft. When
the number of latent states $K$ is further increased, the latent states can be clustered in groups of high, low and medium CAPE
categories. In Fig. A1 top boxplots of CAPE categories by 3 latent states for daily mean (left: day and right: night) and in
the middle the affiliation of CAPE categories to the latent states are presented. For higher $K$, the number of latent states
with affiliation without gaps is higher at night compared to day. In Fig. A3 and A4 we use CAPE as input with a resolution
of $500\,\text{km} \times 500\,\text{km}$ on large scale and a grid of $15\,\text{km} \times 15\,\text{km}$ for the output. The output is on convective scale. Here the
affiliations are without gaps as well. The refined category for the convective scale leads to a difference in the distributions
over the latent states for day and night. For day 4 and for night 5 mean categories are distributed according to the violin plot
of Fig. A3. The corresponding conditional probabilities can be seen in Fig. A4. At night the highest conditional probabilities


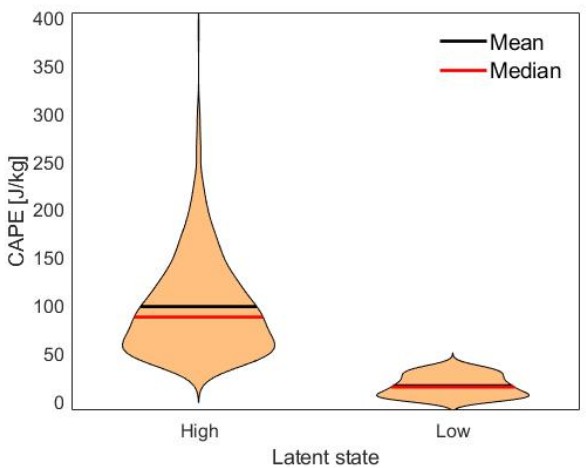

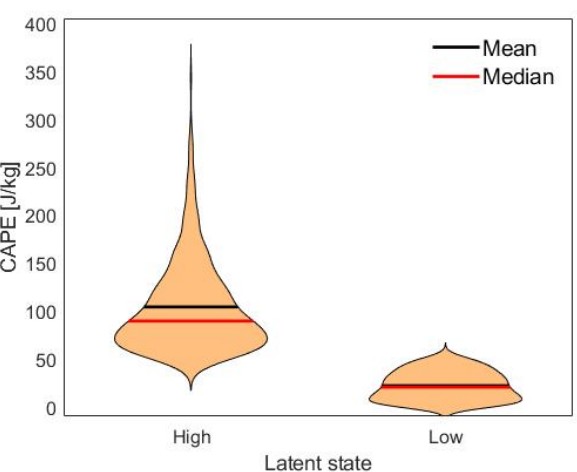

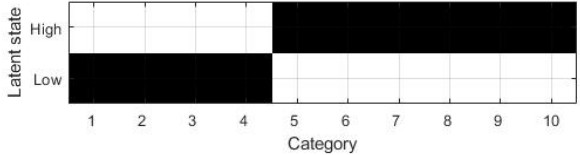

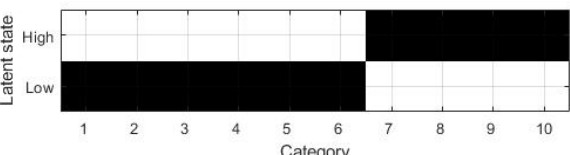

**Figure 7.** Top: Distribution of CAPE (500 km) in terms of latent states based on kernel density estimation; Bottom: Affiliation of CAPE categories to the latent states; Left (Day) and right (Night); maximal CAPE value around 400 J/kg; 15 km step size for vertical velocity (15 km × 15 km km subdomains for vertical velocity)


**Figure 8.** Probabilities of numbers of updrafts (#1) and downdrafts (#3) conditioned on latent states $P[\#1, \#3 | Z]$; Day (left) and night (right) and latent state "High" (top) and latent state "Low" (bottom); $500\,\text{km} \times 500\,\text{km}$ domain for CAPE and $15\,\text{km} \times 15\,\text{km}$ subdomains for vertical velocity.



for high and low CAPE categories appear in the middle box of the axes of number of up- and downdrafts, i.e. high up- and downdraft numbers, low number of boxes with no vertical draft activity. For the mean categories the conditional probability is concentrated at the origin around zero. During daytime, the probabilities are drawn to the diagonal due to the higher variance.

### 4.3 Discussion related to atmospheric dynamics

In Sect. 4.1 we discussed the results of the Bayesian model reduction from a mathematical perspective and in Sect. 4.2 we interpreted the outcomes for a higher number of latent states. The method groups input categories into fewer latent states. These are interpreted as reduced states for the large-scale atmospheric dynamics with respect to their probabilistic impact on vertical motion. We applied an energetic variable as the driver on large scale. CAPE is the convective available potential energy. It does not have to be fully available, meaning that high CAPE values does not *necessarily* lead to convective activity

on smaller scales but increases the probability of smaller scale convective activity. The release of kinetic energy of a certain CAPE level to vertical movement needs triggers such as flows over mountains or forests which lead to instabilities of the hydrostatic equilibrium. The dependence on surface conditions on the earth requires a probabilistic way of thinking. Therefore the mathematical tool DBMR provides a simple probabilistic description. Using the method, we intend to draw conclusions about categorical processes in the atmosphere. Since the system can not be in two different categories simultaneously, categories

are disjoint and the relation between the probability for large scale and smaller scales can be formulated via the conditional probabilities and the conservation of the total probability. The methodology breaks up probability calculations into distinct parts and relates marginal probabilities to conditional probabilities. The aim of this work is to test the stochastic method in an meteorological application towards a reduced categorical model of smaller scale convective activity in the atmosphere depending on large scale drivers.

To analyze the relation of large scale dynamics in the atmosphere to smaller scale categorical processes, the COSMO-REA6 reanalysis data set was applied (Bollmeyer et al., 2015). We averaged CAPE for 500 km × 500 km and the vertical up- and downdrafts in 125 km × 125 km domains, as described in Sect. 3.2. Regarding the summer months July and August in the years 1995 to 2015, CAPE reaches averaged values between 0 and 400 J/kg and the vertical velocities have ranges from -0.15 to 0.2 m/s on mesoscale and -1.7 to 1 m/s on convective scale. In the meteorological setting we showed how the Bayesian model

reduction performs. We combined large-scale CAPE with a subgrid-mesoscale time series for vertical velocity and count the numbers of up- and downdrafts. Therefore we mapped vertical velocities as updraft, no draft and downdraft dependent on an interval around zero vertical velocity. In the preprocessing of Sect. 4.1 we adjusted the interval for vertical draft with range 0.0096 m/s according to the meteorological data. The interval was chosen symmetrically on the basis of the histogram of mean vertical velocities in Fig. 4. We chose a number of 10 input categories and reduced these to two latent states. This was done for

day and night, respectively.

  In Fig. 5 the summary statistics with the affiliation of input categories to the latent states are presented. The affiliations in Figs. 5 and 7 have no gaps, meaning that the affiliations are interrelated and are not interrupted. The affiliations lead to a separation of the latent states in the boxplots for day and night. Thus a certain range of CAPE values can be assigned to every latent states. During daytime the range of values for the latent state "High" is at around 400 J/kg and greater compared to





the corresponding latent state during nighttime. For smaller scales we reduced the number of output categories. In Fig. 7 at
the bottom, 6 high and 4 low CAPE categories for daily mean and 4 high and 6 low CAPE categories at night are affiliated.
As a result of the averaging, the categories are almost evenly distributed over the latent states. The convective activity of the
atmosphere is stronger during the day than during nighttime. Therefore, the vertical draft is less at night than during the day.
Mean and median are around 100 J/kg for the latent state "High" and 25 J/kg for the latent state "Low". The mean and median

are similar for day and night. There is a difference for the variance. At night the distribution of latent state "High" is sharper
due to less variance, only 4 categories are affiliated compared to the daily mean.

     Joint probability distribution of number of grid points with positive and negative vertical velocity conditioned on the resulting
latent states are shown in figs. 6 and 8. The sum of the probabilities of all categories for every box is 1. Increasing the interval
for vertical draft makes less up-/downdrafts, thus moving probability mass away from the diagonal, where large fractions of

up-/downdrafts are sitting. There are higher probabilities during daytime for medium to high numbers of up- or downdraft.
Lots of updrafts during daytime lead to the existence of a lot of downdrafts due to mass conservation. At night due to less
vertical draft, low to medium numbers of up- or downdraft are higher. For the latent state "Low", the distributions in Figs. 6
and 8 concentrate on higher and lower numbers of downdrafts and small numbers of updraft. The representation of probabilities
of numbers of updrafts and downdrafts conditioned on the latent states for the convective scale in Fig. 8 correspond in their

distributions to the results on mesoscale in Fig. 6.

     The generation of kinetic energy of a certain CAPE level to vertical draft on smaller scales can occur up to a few hours later.
A temporal shift for the in- and output could have an effect on the stochastic relation shown in Fig. 6. We consider the 12 hours
means. For data with a higher temporal resolution, one could realize a shift of 2-4 hours for the input. This is deferred to future
studies.

**5  Conclusions**

    It is of importance to identify stochastic models by using categorical approaches compared to fluid mechanics described by
continuous partial differential equations. In this study, a recent algorithmic framework called Direct Bayesian Model Reduction
(DBMR) (Gerber and Horenko, 2017) is applied which provides a scalable probability-preserving identification of reduced
models directly from data. We assume that the output of a Bayesian model depends on the input through a latent variable,

which can merely take a small number of different latent states. The stochastic method is tested in a meteorological application
towards a model reduction to latent states of smaller scale convective activity conditioned on large scale atmospheric flow.

     We combined the convective available potential energy (CAPE) as large scale flow variable with smaller scale subgrids time
series for vertical velocity. Therefore we mapped vertical velocities as updraft, no draft and downdraft dependent on an interval
around zero vertical velocity and count the numbers of up- and downdrafts. Data sets of daily means of 12 hours for day and

night were computed using COSMO-REA6 reanalysis over a domain that covers Germany for a period of the summer months
July and August in the years 1995 to 2015. In the analysis the scales from 500km to 125km (mesoscale) and up to 15km were





considered. The categorical data analysis was done for day and night and discussed for different numbers of latent states. We chose a number of 10 input categories and reduced these to two and three latent states.

The step from the fluid continuum described by partial differential equations to a categorical stochastic description with DBMR provides a reduced model defined on a set of a few latent variables. These are interpreted as reduced states for the large scale atmospheric dynamics with respect to their probabilistic impact on vertical motion. For 2 latent states the input is separated into categories with high and low CAPE values whereas for 3 latent states we have an affiliation to categories with high, medium and low CAPE values. The output categories for the vertical velocity describe the number of up- and downdrafts. In the result, we gain conditional distributions for the numbers of up- and downdrafts conditioned on the latent states for day

and night. In the application we found a probabilistic relation of CAPE and vertical up- and downdraft.

For a resolution of 125km we applied a $4 \times 4$ grid and had 16 boxes with vertical velocities. During daytime the chance for updraft is higher conditioned on the latent state with high CAPE values. Probability adds up for numbers of up- or downdrafts higher than 10 to 81%. The distribution for the latent state with low CAPE values has higher probabilities at high numbers of downdrafts. Here the probability of numbers of downdrafts of 6 to 16 is 68%. At night probability adds up at small numbers

of downdrafts for the latent state with high CAPE values. For low CAPE values, we observe that a low number of updrafts is likely. The probability is accumulated to 82% for the number of updrafts between 0 and 4.

On smaller scale with a resolution of 15km we applied a $32 \times 32$ grid and had 1024 boxes with vertical velocities. We divided the output into 3 categories of low (0 to 341), medium (342 to 683) and high (84 to 1024) numbers of up- and downdrafts. During daytime the probability for a medium number of up- and downdrafts is 34% for the latent state with high CAPE values.

Here low and high numbers of up- and downdraft have small probability. For low CAPE values the maximum in the distribution occurs for a medium number of downdrafts and low number of updrafts at 50%. At night the probability adds up at low to medium numbers of downdrafts for the latent state with high CAPE values and for low CAPE values, we observe that the chance of low and medium number of updrafts is 82%. The distribution for the smaller scale resolution (15km) is a stochastic aggregation of the distribution with resolution of 125km. Therefore the distributions are qualitatively similar. When the number

of latent states is further increased, the latent states can be clustered in groups of high, low and medium CAPE categories.

The model reduction of smaller scale convective activity is part of a development process for a model with a stochastic component for a conceptual description of convection embedded in a deterministic atmospheric flow model. Various energetic variable are applicable on large scale. A potential driver to control small scale models is the Dynamic State Index (DSI) (Müller et al., 2020; Müller and Névir, 2019), an "adiabaticity indicator". Other large scale variables driving the smaller scale

stochastics are the available moisture or vertical wind shear. The presented approach provides a basis for further research of smaller scale convective activity conditioned on other possible large scale drivers.

*Author contributions.* Annette Müller prepared the meteorological data. All authors have then contributed to develop the work and prepared the manusscript.



*Acknowledgements.* This research has been funded by Deutsche Forschungsgemeinschaft (DFG) through grant CRC 1114 'Scaling Cas-
cades in Complex Systems, Project Number 235221301, Project A01 'Coupling a multiscale stochastic precipitation model to large scale
atmospheric flow dynamics'.

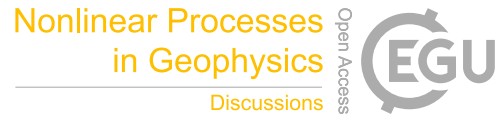

**Appendix A**





**Figure A1.** Top: Boxplot of CAPE categories for 3 latent states "High", "Mean" and "Low" for day (left) and night (right); Middle: Affiliation of CAPE categories; Distribution of CAPE (500 km) in terms of latent states "High", "Mean" and "Low" based on kernel density estimation; $500\,\mathrm{km} \times 500\,\mathrm{km}$ domain for CAPE and $125\,\mathrm{km} \times 125\,\mathrm{km}$ subdomains for vertical velocity.


**Figure A2.** Probabilities of numbers of updrafts (#1) and downdrafts (#3) conditioned on latent states $P[\#1, \#3 | Z]$; Day (left) and night (right) and latent state "High" (top), latent state "Low" (bottom), latent state "Mean" (middle); $500\,\mathrm{km} \times 500\,\mathrm{km}$ domain for CAPE and $125\,\mathrm{km} \times 125\,\mathrm{km}$ subdomains for vertical velocity.



**Figure A3.** Top: Boxplot of CAPE categories for 3 latent states "High", "Mean" and "Low" for day (left) and night (right); Middle: Affiliation of CAPE categories; Distribution of CAPE (500 km) in terms of latent states "High", "Mean" and "Low" based on kernel density estimation; $500\,\mathrm{km} \times 500\,\mathrm{km}$ domain for CAPE and $15\,\mathrm{km} \times 15\,\mathrm{km}$ subdomains for vertical velocity.




**Figure A4.** Probabilities of numbers of updrafts ($\#1$) and downdrafts ($\#3$) conditioned on latent states $P[\#1, \#3 | Z]$; Day (left) and night (right) and latent state "High" (top), latent state "Low" (bottom), latent state "Mean" (middle); $500\,\mathrm{km} \times 500\,\mathrm{km}$ domain for CAPE and $15\,\mathrm{km} \times 15\,\mathrm{km}$ subdomains for vertical velocity.





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
