# Peer review of "Direct Bayesian model reduction of smaller scale convective activity conditioned on large scale dynamics"

_Nonlinear Processes in Geophysics, 2021_

## Referee Comment (RC2)

**Review: Direct Bayesian model reduction of smaller scale convective activity conditioned on large scale dynamics**

**General comments**

The authors investigate the probabilistic impact of large scale atmospheric flow on small scale convective activity, using CAPE and vertical velocity as indicators, respectively. They apply a Direct Bayesian Model Reduction (DBMR) algorithm which was presented previously, to find so called latent states of the categorical input variable – the spatially averaged CAPE (averaged over (500km)² ) and to estimate the probabilities of the categorical output – the number of small scale boxes with upward and downward vertical velocity conditioned on these latent states.

It becomes obvious that the authors intensively studied the topic. However, the manuscript suffers from many impressions and it is sometimes hard to keep track of the computational steps which are applied to data of different spatial and temporal scale.

The manuscript gives the impression, that it has been written in a hurry and I recommend to carefully review the manuscript internally and fix the numerous inaccuracies. Also, the readability should be improved. The captions of the figures are often not clear and the main text should be more to the point.

Unfortunately, the authors do not provide any assessment of the performance of the deduced model, so I find it hard to judge if the presented application of the DBMR yields useful results. Also, it would be nice if the authors could sketch, how the stochastic models deduced in the manuscript could be used in the future. It is mentioned, that there is in general the need for stochastic parametrization, but it is not made explicit, how this study concretely contributes to the issue. What has been achieved by deducing a stochastic model for updraft and downdraft (without any specification of intensity) on the small scale given a large scale value for CAPE?

For the presented study, the raw data must be assigned to categories. For what the authors term the 'categorical input data' the choice of categorization is comprehensible, even though the number of 'causality boxes (n=10)' is introduced ad hoc. For the 'categorical output data' the authors provide little on their choice of categories. Why are the three categories 'updraft', 'downdraft', 'no draft' important? And why does the intensity of the draft not matter in this study?

Regarding the data, as mentioned previously, it is sometimes hard to decipher which data was ultimately used. For example, Fig. 2 shows an area for which data is available, but seemingly all results presented in Sec.4 are solely based on the data from the North-West quadrant in Fig.2, if I understood correctly. Also, it unclear when hourly data plays a role and when if averages are taken before the computation of CAPE or afterwards. Or is CAPE given as a variable in the data set? Please see the specific comments to Sec. 3.

In Sec. 3.2.1 the authors propose two different ways to categorize the input data. However, in the remainder of the manuscript, apparently only one of these is considered – which one is not further specified.

The clearest section is Sec.2 which presents the model. However, this is not the work of this study, but rather a summary of Gerber and Horenko (2017). It remains unclear to me, why the method is considered to be Bayesian.

Is the fact that $\Gamma^*_{\{kj\}} \in \{0, 1\}$ (see l. 114) actually a constraint imposed on the algorithm or something that follows immediately from the definition of the model and the approximation of the log-likelihood given by Eq (9)? If it is a constraint, please elaborate on why this is chosen.

In this context, also the fact that the latent states are found by the algorithm itself, by assigning the $\Gamma^*_{\{kj\}}$ should be emphasized stronger and be revisited in Sec. 4.1.2.

**Specific comments**

l.5     The categorization is based on the conservation of total probability.

         The meaning of the sentence is not very clear at this stage. What is 'the categorization'? And, should the conservation of total probability not be a trivial fact?

l.9     Should it not be 'reasearch on' instead of 'research of'?

l.15    Due to the geostrophic and hydrostatic equilibrium there is a scale separation induced by thermal stratification, gravity and rotation for scales above several kilometers (Klein, 2010).

         Does this mean, that process of scale below several kilometers are separated from those of scale above several kilometers? Or can processes of scale above several kilometers be separated in different groups regarding to their spatial scale? This is a bit unclear.

         Proposal: For processes of spatial scale above several kilometers, geostrophic and hydrostatic equilibria induce a temporal seperation of scales.

l.16    Medium-range forecasts are made up to 10 days in advance.

         Medium range forecasts of what?

l.17    Predictions of convection further in advance cannot be deterministic and are highly uncertain because errors of the initial space of the smaller scales are growing.

         What do you mean by 'errors of the initial space? Do you mean errors in the estimate of the initial state?

         The fact that errors grow is nothing special. Can you say something about the particularity of the error growth that apparently hinders forecasting?

l.19    A new perspective for improving General circulation models (GCMs) came from parameterizations.

         Please clarify what is new. The use of parametrizations itself is not new.

l.21    Nowadays, many data-driven approaches are dwelling on stochastic parametrization methodologies involving the convective available potential energy (CAPE) as large scale driver for convection, e.g. in (Khouider et al., 2010; Dorrestijn et al., 2013a, b).

Is this citation in line with the citation style of the journal?

l.23    Their approaches need high computing capacities, but the costs to process large quantities of data can become a limiting factor.

It seems 'but' is the wrong conjunction here. Please replace by 'and' or clarify.

l.24    The statistical analysis of atmospheric dynamics simulations requires dimensionality reduction techniques which yield applicable reduced models.

This sounds as if ANY statistical analysis of such simulations would rely on dimensionality reduction. I am not certain, since this is not my field of expertise, but I doubt that this is true. Please formulate this statement more carefully.

l.29    The applicability of many approaches is based on the identification of reduced models defined on a small set of latent states.

This is a quite generic sentence. What are 'many approaches' here? Also, doesn't this in parts duplicate the statement

'The statistical analysis of atmospheric dynamics simulations requires dimensionality reduction techniques which yield applicable reduced models.'

from above (l.24)?

l.30    These methods derive aggregations of original variables based on a reduced approximation of the system in terms of relation matrices.

It is still not clear what methods this sentence refers to.

Both sentences, l.29 and l.30 receive meaning from the subsequent sentence. Not optimal, but okay.

l.42    Relation between the probability for large scale and smaller scales can be formulated categorically via a conditional probabilities and the conservation of the total probability.

I thought the large scale process will be treated as a given variable? Also, I think the word 'relation' requires an article here.

l.52    Our aim is to study and understand a stochastic relation between two variables X and Y that can take values from two finite sets.

In which way? Are the sets the same for both variables? Maybe, 'that can take values from two corresponding finite sets each'?

Sec. 2.1 What about the stationarity of the processes? This should be a requirement for the method.

l.128 To apply DBMR, categorical processes for the in- and output have to be defined.

This sounds a little odd. Why would you 'define' processes? I thought the in- and output would be observational data? You do not specify the model of a categorical process, but rather you bin the observational data in order to make it categorical.

l.135 It is defined by Eq.(11) where $\theta_e$ is the pseudopotential temperature of the ascending air parcel, $\theta$ is the potential temperature of the surrounding air, and $z_{LFC}$ is the so-called Level of Free Convection (LFC). The LFC is the height at which the rising air parcel becomes significantly warmer than its environment; $Z_{ET}$ denotes the height, where the rising air parcel has the same temperature as its environment (ET stands for equal temperature). Thus, regarding its definition (11), CAPE becomes large if the temperature difference between the rising air and the environmental air is large, see (Bott, 2016, p. 431 ff).

The situation described here, is a little unclear. I assume $z_{LFZ} < z_{ET}$, correct? Further, we consider a air parcel, that is located at $z_{LFZ}$ and that is about to rise, or one that is at height $Z_{ET}$? Or is this irrelevant? What is 'significantly warmer than its environment'? I assume $\theta$ is a function of z? Otherwise, to which height does $\theta$ refer? What about $\theta_e$? Is this a function of z? Maybe a little figure could be helpful.

l.151 To analyze the relation of large and small scale parameters, the COSMO-REA6 reanalysis data set is used (Bollmeyer et al., 2015).

This is very generic, you do not investigate the relation between small and large scale parameters in principle.

l.154 Since we focus on smaller scale convective events conditioned on large scale dynamics in the atmosphere, we consider the summer months July and August in the years 1995 to 2015.

Why? Because these months feature most small scale convective events? Why do you neglect the rest of the time series?

l.157 This subdomain is bounded by the $45.2\circ N - 54.7\circ N$ , $5.8\circ E - 15.3\circ E$ and shown in Fig. 2.

I wonder if 'bounded by **the** [coordinates]' is a correct way to express an area.

Please, do not use the word 'subdomain' here, use 'domain' instead. The subdomains will be introduced at a later stage.

l.158  The Northwest coordinate is $(5.8\circ E; 54.7\circ N )$ and the Southeast coordinate is $(15.3\circ E; 45.2\circ N )$.

This is trivial and can be deleted.

Figure 2. Why are 500km x 500km quadrants so distinctly rectangular and not quadratic? Please check 'domain' wording.

l.162     The domain that covers Germany in Fig. 2 is divided into four 500 km × 500 km quadrants, where the spatial arithmetic mean of each of the quadrant is considered such that we obtain one CAPE value for each quadrant.

Do you average over the atmospheric variables provided by COSMO-REA6 and compute one value for CAPE subsequently? Or do you first compute a spatially dependent CAPE and average afterwards?

l.164     We separate and filter the data of CAPE and the vertical velocity in further subdomains in order to define the categorical in- and output.

What is meant by 'separate and filter' in this context?

After reading 3.2 the most likely interpretation seems to me, that for CAPE you use averages over the four quadrants, while for the vertical velocity you consider spatial averages over boxes of different sizes. But this is not clear from the text, especially since you write 'we separate the data of CAPE and the vertical velocity in further subdomains'.

l.165     The corresponding sizes of the subdomains are summarized in Tab. 1.

Now the wording really becomes a mess. First, 'subdomain' is used for the entire region that covers Germany in l. 158. Second, it is used to refer what is called 'quadrants' in the text (l.162) and here it is used for even smaller 'subdomains'?

l.166     For the analysis with DBMR, the northwest quadrant 1 over Holland in Fig. 2 is used.

Does this mean the rest of the data is not considered any further? Application of DBMR is the main purpose of this study as far as I understand. Then why do you introduce the entire data?

l.167     There is no influence of the Alps on smaller scale convective activity.

Does this refer to the 1 quadrant or to all quadrants?

l.197     There are exactly $(m + 1)^2$ ways to decompose m into the (ordered) sum of 3 nonnegative numbers.

I would not call this a decomposition. Rather, I would say, there are $(m+1)^2$ possibility to assign m observations to 3 different categories.

Why should the sum be ordered? If you consider an ordered assignment, the number should be larger.

Finally, I think you are missing a factor of ½. Please check.

l.198     In our probability-preserving algorithm the number of the occurring categories in the data are counted for the categorical observational input and output. The probability of a category is estimated by its occurrence frequency with respect to the total number of data points.

What 'algorithm' does this statement refer to? Apparently not the DBMR, since this algorithm relies on counting occurences. Does that mean that you compute Eq.(6) explicitly? If so, please explain why this is possible. In the beginning, you motivate the use of DBMR by saying, that this is often times computationally very costly.

Also, why does solving Eq.(6) require an 'algorithm'?

l.207   We also evaluate the exact log-likelihood, as in (5). Fig. 3 shows the exact in blue and the relaxed log-likelihood in red, both for the reduced problem, i.e., the one with latent states.

Equation (5) to my understanding refers to the case without latent states. \Lambda in Eq.(5) must be replaced by \lambda * \Gamma to compute the likelihood of the reduced model.

l.209   The only parameter in the algorithmic procedure introduced above is the reduced process dimension K for the number of collective **causality boxes**.

You have not used the term 'causality boxes' before. Please introduce properly.

Also, what about 'm', the number of subdomains?

By the way, 'm' might not be the best choice for the number of subdomains, since in Sec. 'm' was introduced as the number of categories for the output variable.

Figure 3 – Caption:    Again, Eq.(5) refers to the full model without latent states. Also, it is not clear what subdomains have been chosen to generate the results presented in the Figure.

Sec. 4.1.1

I do not understand, why the content of this section is presented under 'results' and not in the data pre- and postprocessing section. I think this would nicely fit into into 3.2.2, where the classification of $w_i$ values is discussed in first place.

The histogram presented in Fig. 4 is not really a result of this work, but can legitimately be considered as part of preprocessing.

l.216   In the following, the pre- and postprocessing on DBMR with respect to the categorical in- and output will be discussed.

Why is this still 'pre-and postprocessing' if it is presented in the 'Resuts' section?

Also, what means 'pre-and postprocessing **on** DBMR'?

l.218   All-day mean data serve as basis for determining the interval for vertical draft.

All-day mean data serve as **a** basis for determining the interval for vertical draft.

However, 'the interval' is an interval for the value of $w_i$ that is classified as 'no draft'. In the next sentence you refer to a 'subclassification'. To me it is not quite clear, what exactly is being subclassified 'by the interval'.

If possible, please reformulate in a more precise manner.

Fig.4  What exactly is shown here? In Sec. 3.1 you write that you use 12h averaged data, but the in caption it says hourly averaged data? Does the word 'mean' in 'mean vertical velocities' refer to a spatial mean over 125km? The data resolution is 6km! 'vertical velocities for day and night' contradicts 'hourly averaged values'.

Please be more precise!

Red vertical lines represent a tube for **NO** vertical draft.

The summer months July and August in the years 1995 to 2015.

In captions it might be acceptable to write incomplete sentences, but this 'statement' is not convenient.

The sample size of the reanalysis data set sums up to S = 1302 (2 × 31 × 21).

Yes, but that is not what you are showing. Why do you actually consider hourly resolved data to define a classification for the 12h averaged data?

Also, you do not give any justification for the choice of the 'no draft' – interval. Please elaborate on the criteria that you apply. Is it a certain percentile in the data?

l.225  On each box, the central mark indicates the median, and the bottom and top edges of the box indicate the 25th and 75th percentiles of CAPE concerning the affiliated n categories (n = 10 divided by quantiles).

What is meant by 'concerning the affiliated n categories (n = 10 divided by quantiles).'

l.227  In Fig. 5 one sees that the input categorization is similar in terms of value for day and night. The first latent state includes 5 (for day) and 4 (at night) CAPE categories with high values. This represents high CAPE values and is therefore referred to as "High". Five (for day day) and 6 (at night) categories are affiliated to the second latent state, which is denoted with "Low".

What criterion are these assignments based on? How are the latent states defined?

I think I understand this better having read the entire manuscript. However, the fact that first a manual categorization is performed and that the DBMR performs a second categorization might be the cause of some confusion here. This could be avoided if you pointed out these two levels of categorization at an earlier stage.

Figure 5. Top: Boxplot of CAPE categories by 2 latent states for daily mean (left: day and right: night); Bottom: Affiliation of CAPE categories to the latent states; Data on large scale: 500 km × 500

km subdomains for northwest of Germany, hourly averaged CAPE; On mesoscale: 125 km × 125 km subdomains for vertical velocity. Time series: 21 years for July and August, S = 1302 (Length of time series)

I understand that in the top two panels, the boxplots show the 12h averaged CAPE data (spatially averaged over the Northwest quadrant of the COSMO-REA6 data) which is assigned to the latent states 'high' and 'low' on the left and right of each of the two top panels, respectively. The left panel shows the 'day' data and the right panel the 'night' data. However, this interpretation contradicts the fact, that the distributions shown for the 'high' and 'low' latent states do in fact overlap. Please clarify, what distributions are shown here?

This is not quite clear from the caption. Please use (a), (b) … to unambiguously mark the panels.

What role plays 'hourly averaged CAPE' in this plot?

What do the red crosses above the 75% quantile of the box plots mean? Are these extreme events? Why are there none below the 25% quantile?

What does 'On mesoscale: 125 km × 125 km subdomains for vertical velocity.' mean in the context of this plot? No vertical velocity data is shown here.

l.233    The difference between the scales is small (375 km) with 500 km step size on large scale and 125 km step size on the smaller scale. The scale jump is of factor 4 on the basis of the small scale.

This is unclear. I understand that you somehow want to relate a (125km)^2 averaged vertical velocity data set with the (500km)² averaged CAPE data. But the (125km)²  averaged vertical velocity data was not specifically introduced before? You are still at the stage, where you define the latent states for the categorical input data, why would you discuss specific choices for the output data resolution, here?

Please clarify at the beginning of the section, that a) you now move to the application of the DBMR algorithm and b) that this application requires a choice of the scale of the categorical output or in other words, a choice of the spatial average taken on the vertical velocity data. Please clarify that for the first DBMR application the spatial scale of the output is set to 125km.

Also, it would help to emphasize, that the choice of the spatial scale for the categorical output will influence the latent states identified by the DBMR.

l.245    In Fig. 6, K bivariate histograms are shown for day and night respectively.

This is a specific case where apparently K=2, so please indicate.

Figure 6: Please use a discretized colorbar. Also check if other colorbars facilitate a better distinction between the many low values. Please use (a), (b) ,… to identify panels

To which quadrant of the COSMO-REA6 data do the results refer to?

l. 256 In the top left panel (Z 1 , day) of Fig. 6 probability adds up for numbers of up- or downdrafts higher than 10 to 81%. In the top right panel, probability accumulates at small numbers of boxes with downdraft.

Judging from an optical assessment there should be at least as much probability weight allocated to states where both, the number of downdraft cells and the number of updraft cells are below 10. This contradicts the above statement. Please check.

l.257 In the top right panel, probability accumulates at small numbers of boxes with downdraft.

I cannot see any of the states with pronounced downdraft having high probability! Much of the probability mass is allocated to states with no downdraft, and little updraft.

l.262 In the bottom left panel (Z 2 , night) the probability is accumulated to 82% for the number of updrafts between 0 and 4.

That's the bottom right panel.

Figure 7: What is the difference between Fig.5 and Fig.7 ? If it is, that you use CAPE values averaged spatially averaged over (125km)² in fig.5 and (15km)² in fig 7, then this is not clear and also contradicts the statements in sec. 3:

The domain that covers Germany in Fig. 2 is divided into four 500 km × 500 km quadrants, where the spatial arithmetic mean
of each of the quadrant is considered such that we obtain one CAPE value for each quadrant.

We use the average of the 500 km × 500 km
quadrants, considering CAPE as the large scale atmospheric driver.

l.270 KDE is a non-parametric way to estimate the probability density function of a random variable.

A KDE requires the choice of a bandwidth and is therefore not non-parametric.

l.283 Affiliations without gaps lead to a separation of the latent states. "No gaps" means that affiliations are interrelated and not interrupted in the middle plots of Fig. A1 and Fig. A3. The affiliations have no gaps for day and night.

Of course the latent states should cover all categorical input states and should have 'no gaps'. Or does 'no gaps' mean that there is an overlap of spatially smaller scale CAPE values which are assigned to the different latent states based on the large scale CAPE average?

l.343 The representation of probabilities of numbers of updrafts and downdrafts conditioned on the latent states for the convective scale in Fig. 8 correspond in their distributions to the results on mesoscale in Fig. 6.

'for the convective scale' should refer to 'probabilities of numbers of updrafts and downdraft', right? In the current version of the sentence, 'for the convective scale' seemingly refers to 'the latent state'.

l.355 The stochastic method is tested in a meteorological application towards a model reduction to latent states of smaller scale convective activity conditioned on large scale atmopsheric flow.

'latent states of smaller scale convective activity' suggests that the latent states are introduce for the vertical velocity, however, they have been introduced for the caterogical input, which is the large scale atmospheric flow, that is CAPE.

**Technical comments**

l.26 Since its introduction by Lorenz (1956), EOF analysis—known as principal component analysis (PCA) or proper orthogonal decomposition (POD)—has become an important statistical tool in atmosphere science.

- also known … -

l.37 The latter approach does not require a distributional assumption but works instead with a discretized state vector.

Why 'the latter'?

l.43 Various energetic variable are applicable on large scale.
variables is missing an s.

l.48 In Sect. 4 the results are discussed related to atmospheric dynamics.

… the results are discussed **and** related to

or the results are discussed with regard to atmospheric dynamics.

l.114 Moreover, the method yields $\Gamma^*\_{kj} \in \{0, 1\}$, i.e., the original input categories are assigned to the reduced system's (latent) categories in a deterministic fashion (no "fuzzyness" in the affiliations).

Why is the $\Gamma^*\_{kj}$ marked with an *?

l.120 This manifests in the variance of the estimated parameter $\lambda_{*ik}$, which shows a K/n-times smaller uncertainty than $\Lambda_{ij}$…

I am not sure the use of 'manifest' is correct here.

l.155 The sample size of the reanalysis data set used in Sect. 2 sums up to $S = 1302$ $(2 \times 31 \times 21)$

Sect. 2? Wrong reference?

l.168 According to the meteorological data in Sect. 3.1…

According to the meteorological data **described** in Sect. 3.1…

l.169   CAPE plays the role of an input variable X in Sect. 2

CAPE plays the role of an input variable X **as defined** in Sect. 2

l.175   With this type of classification, extreme weather events tend to be in a separate category.

tend to **be in separate categories**.

l.175   These are not Gaussian distributed.

Grammatically the reference of 'these' is not clear. Could refer to 'categories' from the previous sentence. I assume you meant to refer to 'extreme weather events'. Please clarify and add 'in terms of CAPE'.

l.193   Let $Y\_i$ (t) be the discretized vertical velocities at time t with $1 \leq i \leq m$ numbering the grid boxes on the corresponding scale, see Tab. ??.

Probably, it is better to not use m as an upper bound for i, since m was already use for the number of observations of Y in time. Here, m refers to the spatial number of observations that are being made simultaneously, if I understand correctly.

Please correct the reference.

l.206   for every fixed number K of latent state.

for every fixed number K of latent state**s**.

l.206   For the respective latent state...

For each K.

'The latent state' would refer to an individual state.

l.220   In Fig. 4, the histogram of mean vertical velocities for a resolution of 125km with the interval is shown.

In Fig. 4, the histogram of mean vertical velocities for a resolution of 125km is shown together with the interval that defines the 'no draft' category.

l.229   Five (for day day) and 6 (at night) categories

double day

l.244   In order to visualize the probabilities of the small scale conditioned on the latent states…

In order to visualize the probabilities of the small scale variable conditioned on the latent states of the large scale variable…

l.244   the entries of the $\hat{\lambda}$

why does \lambda carry a hat?

---

## Author Response (AR1)

**Response to the Reviewer #1 and #2 "Direct Bayesian model reduction of smaller scale convective activity conditioned on large scale dynamics"**

Robert M. Polzin, Annette Müller,

Henning W. Rust, Peter Névir, Péter Koltai

December 7, 2021

Dear Editor and Reviewers,

thank you for the comprehensive and helpful evaluation of our manuscript. We are carefully revising the manuscript considering all of your comments. Below you will find the general comments, specific comments and technical comments and our **answers**. We refer to all **changes** in the revised manuscript.

**Contents**

**1 Referee 1 (R1)**

**1.1 Comments (C)**

1. The study uses the Direct Bayesian Model Reduction method. However, I did not get the impression that a Bayesian approach has been taken since the parameters

have been estimated using maximum likelihood. Perhaps I have missed where the Bayesian aspect is coming in. Can the authors explain this in more detail.

**Change:**

Pages 3-4

**Answer:**

In DBMR, „Bayesian" refers to the fact that the estimated model is a „Bayesian relation model" (incorporating distributions of input, output, and their conditional probabilities) and not to the likelihood maximization in the computations. In this sense, the title for section 2.2 was ambiguous, and we thank the reviewer for pointing this out. We changed subsection 2.1 to „Full Bayesian relation model" and 2.2 to "Maximum likelihood approach".

2. Eq. (12): Why don't you use the standard "w" for the vertical velocity?

**Change:**

**Answer:**

$w$ ($z$-Sytem) or $\omega$ ($P$-system) are standard for the vertical velocity. Eq. (12) was rewritten.

3. The authors should check the citation format. For example, "In (Kirkpatrick et al., 2009)" should read "In Kirkpatrick et al. (2009)".

**Change:**

Citation format in the manuscript

**Answer:**

We changed the citation format in the manuscript, e.g. to "In Kirkpatrick et al. (2009)".

4. Can the authors comment on how they plan to deal with uncertainties and biases in vertical velocity for parameterizations. The vertical velocity can be hard to measure and is likely biased in reanalysis and model data.

**Change:**

**Answer:**

Indeed, vertical velocity is likely to be biased and uncertain in reanalyses. Here, we worked with discretize vertical velocity and thus with a less precise variable. This makes the problem of uncertainty and bias less relevant but is definitively not a relief. In a stochastic model for the updraft which is to be developed, one can think of including an additional parameter as factor to the vertical velocity to allow for a tuning with respect to the effect generated by the modelled updraft.

5. Line 194: Correct "Tab. ??."

**Change:**

**Answer:**

We fixed the typo.

**2 Referee 2 (R2)**

**2.1 General comments (GC)**

1. It becomes obvious that the authors intensively studied the topic. However, the manuscript suffers from many impressions and it is sometimes hard to keep track of the computational steps which are applied to data of different spatial and temporal scale.

   **Changes:**

   **Answer:**

   We added the necessary explanations and restructuring to clarify when and why which data is used for the analysis. Raw data are hourly REA6 data. We first computed CAPE as REA6 variable and then averaged for the respective spatial scale to 12 hours means.

2. The manuscript gives the impression, that it has been written in a hurry and I recommend to carefully review the manuscript internally and fix the numerous inaccuracies. Also, the readability should be improved. The captions of the figures are often not clear and the main text should be more to the point.

**Changes:**

Table of contents, captions, main text

**Answer:**

We reconsiderd the structure with regard to a clear division of the pre- and postprocessing and the results and discussion. We moved the content regarding the interval for vertical draft of subsection 4.1 labeled "Results" to the subsection 3.2.3. We took a lot of time to revise the captions of the figures and and the main text.

3. Unfortunately, the authors do not provide any assessment of the performance of the deduced model, so I find it hard to judge if the presented application of the DBMR yields useful results. Also, it would be nice if the authors could sketch, how the stochastic models deduced in the manuscript could be used in the future. It is mentioned, that there is in general the need for stochastic parametrization, but it is not made explicit, how this study concretely contributes to the issue. What has been achieved by deducing a stochastic model for updraft and downdraft (without any specification of intensity) on the small scale given a large scale value for CAPE?

**Change:**

Pages 10-11

**Answer:**

Although quantification of model performance is possible here using, e.g., a cross validation study given an adequate score of interest, it is probably not very helpful at this stage. We considered our study rather as a proof-of-concept ideally preparing grounds for a stochastic model for vertical movement to be inserted into a circulation model. Usefulness should be evaluated then in terms of circulation model simulations. We emphasized the nature of this study in Sect. 3.3 to make this more clear. Further work is required to give the latent states a meteorological meaning (in the sense of circulation weather types, regarding all seasons separately).

4. For the presented study, the raw data must be assigned to categories. For what the authors term the 'categorical input data' the choice of categorization is comprehensible, even though the number of 'causality boxes (n=10)' is introduced ad hoc. For the 'categorical output data' the authors provide little on their choice of categories. Why are the three categories 'updraft', 'downdraft', 'no draft' important? And why does the intensity of the draft not matter in this study?

**Change:**

Pages 8-9

**Answer:**

The chosen categorization depends on the amount of available data. This holds for the 10 input categories, as well as for the three output categories. However, within a certain range, we varied input and output category numbers and judged by subjective physical plausibility. We recognized that there are other probably more important factors to vary, such as the linear or quantile categorization, the driving variables on the large scale or the scale of input and output variables. The predefinition of 'updraft', 'downdraft', 'no draft' determines whether there is convection and, if so, how it is directed (upwards or possibly downwards).

5. Regarding the data, as mentioned previously, it is sometimes hard to decipher which data was ultimately used. For example, Fig. 2 shows an area for which data is available, but seemingly all results presented in Sec.4 are solely based on the data from the North-West quadrant in Fig.2, if I understood correctly. Also, it is unclear when hourly data plays a role and when if averages are taken before the computation of CAPE or afterwards. Or is CAPE given as a variable in the data set? Please see the specific comments to Sec. 3.

**Change:**

**Answer:**

We clarified this in a revised version of the manuscript. 12h mean values based on hourly data are used. First we calculated CAPE and then average CAPE values. Moreover, data is available for Germany. In order to focus our method we started at the top left with the first quadrant (see Fig. 2). Here we expected the relatively flat surface in the north of Germany to be more homogeneous and different from the pre-alpine southern regions with forced uplifting.

6. In Sec. 3.2.1 the authors propose two different ways to categorize the input data. However, in the remainder of the manuscript, apparently only one of these is considered – which one is not further specified.

**Changes:**

**Answer:**

We included the classification into quantiles.

7. The clearest section is Sec.2 which presents the model. However, this is not the work of this study, but rather a summary of Gerber and Horenko (2017). It remains unclear to me, why the method is considered to be Bayesian.

**Change:**

**Answer:**

In DBMR, „Bayesian" refers to the fact that the estimated model is a „Bayesian relation model" (incorporating distributions of input, output, and their conditional probabilities) and not to the likelihood maximization in the computations. In this sense, the title for section 2.2 was ambiguous, and we thank the reviewer for pointing this out. We changed subsection 2.1 to „Full Bayesian relation model" and 2.2 to "Maximum likelihood approach".

8. Is the fact that $\Gamma_{kj}^* \in \{0,1\}$ (see line 114) actually a constraint imposed on the algorithm or something that follows immediately from the definition of the model and the approximation of the log-likelihood given by Eq. (9)? If it is a constraint, please elaborate on why this is chosen.

**Change:**

**Answer:**

The binary nature of $\Gamma^*$ is a property of the optimal solution, as it is shown by Gerber and Horenko. The optimal assignment of projector elements resulting from the DBMR-algorithm appears to be either zero or one which requires much less basis functions. We clarified this in the revised version.

9. In this context, also the fact that the latent states are found by the algorithm itself, by assigning the $\Gamma_{kj}^*$ should be emphasized stronger and be revisited in Sec. 4.1.2.

**Change:**

**Answer:**

The structure of $\Gamma^*$ is assigned by minimizing information entropy (likelihood bound). We emphasized stronger that the latent states are found by the algorithm itself.

**2.2 Specific comments (SC)**

1. l.5 The categorization is based on the conservation of total probability. The meaning of the sentence is not very clear at this stage. What is 'the categorization'? And, should the conservation of total probability not be a trivial fact?

**Change:**

**Answer:**

We have changed both sentences. A Bayesian relation model between categorical processes is used. This can be formulated via the conditional probabilities and the law of the total probability.

2. l.9 Should it not be 'research on' instead of 'research of'?

**Change:**

**Answer:**

We have changed 'research of' to 'research on'.

3. l.15 Due to the geostrophic and hydrostatic equilibrium there is a scale separation induced by thermal stratification, gravity and rotation for scales above several kilometers (Klein, 2010).

Does this mean, that process of scale below several kilometers are separated from those of scale above several kilometers? Or can processes of scale above several kilometers be separated in different groups regarding to their spatial scale? This is a bit unclear.

Proposal (from referee): For processes of spatial scale above several kilometers, geostrophic and hydrostatic equilibria induce a temporal seperation of scales.

**Change:**

**Answer:**

We followed your proposal and changed the sentence to: "For processes of spatial scale above several kilometers, geostrophic and hydrostatic equilibria induce a spatial-temporal separation of scales." Space and time are coupled in terms of predictability. The constancy of energy dissipation includes a functional dependency between the spatial and the time scales.

4. l.16 Medium-range forecasts are made up to 10 days in advance.

Medium range forecasts of what?

**Change:**

**Answer:**

5. l.17 Predictions of convection further in advance cannot be deterministic and are highly uncertain because errors of the initial space of the smaller scales are growing.

   What do you mean by 'errors of the initial space? Do you mean errors in the estimate of the initial state?

   The fact that errors grow is nothing special. Can you say something about the particularity of the error growth that apparently hinders forecasting?

   **Change:**

   **Answer:**

   We have changed the sentence to: "Predictions of convection further in advance cannot be deterministic and are highly uncertain because errors of the variable on small scale at the initial state are growing."

6. l.19 A new perspective for improving General circulation models (GCMs) came from parameterizations.

   Please clarify what is new. The use of parametrizations itself is not new.

   **Change:**

   **Answer:**

   The use of stochastic parameterizations are new. We have changed the paragraph and only mention stochastic parameterizations.

7. l.21 Nowadays, many data-driven approaches are dwelling on stochastic parametrization methodologies involving the convective available potential energy (CAPE) as large scale driver for convection, e.g. in (Khouider et al., 2010; Dorrestijn et al., 2013a, b).

   Is this citation in line with the citation style of the journal?

   **Change:**

   Citation style in manuscript

   **Answer:**

   We changed citation style throughout the manuscript.

8. l.23 Their approaches need high computing capacities, but the costs to process large quantities of data can become a limiting factor.

It seems 'but' is the wrong conjunction here. Please replace by 'and' or clarify

**Change:**

**Answer:**

We have changed the sentence to: "Their approaches need high computing capacities and the costs to process large quantities of data can become a limiting factor."

9. l.24 The statistical analysis of atmospheric dynamics simulations requires dimensionality reduction techniques which yield applicable reduced models.

This sounds as if ANY statistical analysis of such simulations would rely on dimensionality reduction. I am not certain, since this is not my field of expertise, but I doubt that this is true. Please formulate this statement more carefully.

**Change:**

**Answer:**

We have rewritten the sentence: "Some statistical analyses of atmospheric dynamics simulations requires dimensionality reduction techniques which yield applicable reduced models."

10. l.29 The applicability of many approaches is based on the identification of reduced models defined on a small set of latent states.

This is a quite generic sentence. What are 'many approaches' here? Also, doesn't this in parts duplicate the statement

'The statistical analysis of atmospheric dynamics simulations requires dimensionality reduction techniques which yield applicable reduced models.'

from above (l.24)?

**Change:**

**Answer:**

We have deleted this sentence.

11. l.30 These methods derive aggregations of original variables based on a reduced approximation of the system in terms of relation matrices.

It is still not clear what methods this sentence refers to.

Both sentences, l.29 and l.30 receive meaning from the subsequent sentence. Not optimal, but okay.

**Change:**

**Answer:**

We have changed the sentence to: "Other examples for reduced approximation in terms of relation matrices are ...; see Gerber and Horenko (2017)."

12. l.42 Relation between the probability for large scale and smaller scales can be formulated categorically via a conditional probabilities and the conservation of the total probability.

I thought the large scale process will be treated as a given variable? Also, I think the word 'relation' requires an article here.

**Change:**

**Answer:**

We changed the sentence to "The Bayesian relation model between large scale and smaller scales can be formulated categorically via a conditional probabilities in the law of total probability."

13. l.52 Our aim is to study and understand a stochastic relation between two variables X and Y that can take values from two finite sets.

In which way? Are the sets the same for both variables? Maybe, 'that can take values from two corresponding finite sets each'?

Sec. 2.1 What about the stationarity of the processes? This should be a requirement for the method.

**Change:**

**Answer:**

As for the first question, these sets can be different, as we discuss in our applications. As for the second question, indeed, we assume that the probabilistic dependence of $Y$ on $X$ is time-independent. Whether $X_t$ and $Y_t$ as $t$-parametrized stochastic processes are themselves stationary does not play a role here. All of this we clarified now.

14. l.128 To apply DBMR, categorical processes for the in- and output have to be defined.

    This sounds a little odd. Why would you 'define' processes? I thought the in- and output would be observational data? You do not specify the model of a categorical process, but rather you bin the observational data in order to make it categorical.

    **Change:**

    **Answer:**

    We have changed the sentence to: "To apply DBMR, a quantization of the input and output processes into categories has to be performed."

15. l.135 It is defined by

    $$CAPE = g \int_{z_{\mathrm{LFC}}}^{z_{\mathrm{ET}}} \frac{\theta_e - \theta}{\theta} \, dz \, , \tag{1} \quad \texttt{\{eq:CAPE\}}$$

    where $\theta_e$ is the pseudopotential temperature of the ascending air parcel, $\theta$ is the potential temperature of the surrounding air, and $z_{\mathrm{LFC}}$ is the so-called *Level of Free Convection* (LFC). The LFC is the height at which the rising air parcel becomes significantly warmer than its environment; $z_{\mathrm{ET}}$ denotes the height, where the rising air parcel has the same temperature as its environment (ET stands for equal temperature). Thus, regarding its definition (1), CAPE becomes large if the temperature difference between the rising air and the environmental air is large.

    The situation described here, is a little unclear. I assume $z_{\mathrm{LFC}} < z_{\mathrm{ET}}$, correct? Further, we consider a air parcel, that is located at $z_{\mathrm{LFC}}$ and that is about to rise, or one that is at height $z_{\mathrm{ET}}$? Or is this irrelevant? What is "significantly warmer than its environment"? I assume $\theta$ is a function of $z$? Otherwise, to which height does $\theta$ refer? What about $\theta_e$? Is this a function of $z$? Maybe a little figure could be helpful.

    **Change:**

    **Answer:**

    For positive CAPE, the difference must be positive. CAPE is determined by the layer thickness between the starting and ending points in space (height) and by the integrand. Boundary conditions can vary. $\theta$ can be a function of $z$, it depends on the difference between the heights and the potential temperature.

16. l.151 To analyze the relation of large and small scale parameters, the COSMO-REA6 reanalysis data set is used (Bollmeyer et al., 2015).

This is very generic, you do not investigate the relation between small and large scale parameters in principle.

**Change:**

**Answer:**

For our studies, the COSMO-REA6 reanalysis data set is used; see (Bollmeyer et al., 2015)."

17. l.154 Since we focus on smaller scale convective events conditioned on large scale dynamics in the atmosphere, we consider the summer months July and August in the years 1995 to 2015.

Why? Because these months feature most small scale convective events? Why do you neglect the rest of the time series?

**Change:**

**Answer:**

The summer months are predestinated for convective events. The months from May to August are possible. We only took two months in order not to have too much data.

18. l.157 This subdomain is bounded by the ... in Fig. 2.

I wonder if 'bounded by the [coordinates] is a correct way to express an area.

Please, do not use the word 'subdomain' here, use 'domain' instead. The subdomains will be introduced at a later stage.

**Change:**

**Answer:**

We used 'domain' instead of 'subdomain'. For the coordinates we wrote: domain $(45.2°N$ to $54.7°N$, $5.8°E$ to $15.3°E)$.

19. l.158 The Northwest coordinate is ... and the Southeast coordinate is ....

This is trivial and can be deleted.

Figure 2. Why are 500km x 500km quadrants so distinctly rectangular and not quadratic? Please check 'domain' wording.

**Change:**

**Answer:**

We erased the redundant information. The reason for the not quadratic representation is the cartographic projection (Mercator).

20. l.162 The domain that covers Germany in Fig. 2 is divided into four 500 km times 500 km quadrants, where the spatial arithmetic mean of each of the quadrant is considered such that we obtain one CAPE value for each quadrant.

   Do you average over the atmospheric variables provided by COSMO-REA6 and compute one value for CAPE subsequently? Or do you first compute a spatially dependent CAPE and average afterwards?

   **Change:**

   **Answer:**

   We first computed a spatially dependent CAPE and averaged afterwards. See R2GC1.

21. l.164 We separate and filter the data of CAPE and the vertical velocity in further subdomains in order to define the categorical in- and output.

   What is meant by 'separate and filter' in this context?

   After reading 3.2 the most likely interpretation seems to me, that for CAPE you use averages over the four quadrants, while for the vertical velocity you consider spatial averages over boxes of different sizes. But this is not clear from the text, especially since you write 'we separate the data of CAPE and the vertical velocity in further subdomains'.

   **Change:**

   **Answer:**

   We have deleted the sentence " We separate and filter the data of CAPE and the vertical velocity in furthersubdomains in order to define the categorical in- and output."

22. l.165 The corresponding sizes of the subdomains are summarized in Tab. 1.

   Now the wording really becomes a mess. First, 'subdomain' is used for the entire region that covers Germany in l. 158. Second, it is used to refer what is called 'quadrants' in the text (l.162) and here it is used for even smaller 'subdomains'?

**Change:**

**Answer:**

We used "domain" for the total region we considered, "quadrant" for the respective quarters of it, and "grid boxes" for the other partitions of the domain.

23. l.166 For the analysis with DBMR, the northwest quadrant 1 over Holland in Fig. 2 is used.

Does this mean the rest of the data is not considered any further? Application of DBMR is the main purpose of this study as far as I understand. Then why do you introduce the entire data?

**Change:**

**Answer:**

We wanted to use all of Germany for the presentation of the data set in Fig. 2. We started at the top left, we expected the relatively flat surface in the north of Germany to be more homogeneous and different from the pre-alpine southern regions with forced uplifting.

24. l.167 There is no influence of the Alps on smaller scale convective activity.

Does this refer to the 1 quadrant or to all quadrants?

**Change:**

**Answer:**

We deleted this sentence.

25. l.197 There are exactly $(m + 1)^2$ ways to decompose m into the (ordered) sum of 3 nonnegative numbers.

I would not call this a decomposition. Rather, I would say, there are $(m + 1)^2$ possibility to assign m observations to 3 different categories.

Why should the sum be ordered? If you consider an ordered assignment, the number should be larger.

Finally, I think you are missing a factor of 0.5. Please check.

**Change:**

**Answer:**

We added explanations to avoid ambiguity here. The assignment of $a_m$ (ordered) observations to three categories can be done in $3^{a_m}$ ways, indeed, but our output $\hat{Y}$ is a three-component non-negative integer vector giving the number of grid boxes that show updraft/no draft/downdraft, respectively. This way, the components of this vector sum to $a_m$ (the number of grid boxes), and the set of all such vectors has cardinality $(a_m + 1)^2$.

26. l.198 In our probability-preserving algorithm the number of the occurring categories in the data are counted for the categorical observational input and output. The probability of a category is estimated by its occurrence frequency with respect to the total number of data points.

    What 'algorithm' does this statement refer to? Apparently not the DBMR, since this algorithm relies on counting occurences. Does that mean that you compute Eq.(6) explicitly? If so, please explain why this is possible. In the beginning, you motivate the use of DBMR by saying, that this is often times computationally very costly.

    Also, why does solving Eq.(6) require an 'algorithm'?

    **Change:**

    **Answer:**

    We changed the sentence to: "In DBMR, the numbers of actually occurring categories are counted for the input and output. These numbers have impact on the probability distribution of the categories for input and output."
    Wording was misleading: the "modeling approach" by a stochastic matrix is probability-preserving, but the algorithm is only to compute the parameters of the model. We call DBMR a modeling approach, and the algorithm is the alternating maximization that computes the two matrices $\lambda, \Gamma$.

27. l.207 We also evaluate the exact log-likelihood, as in (5). Fig. 3 shows the exact in blue and the relaxed log-likelihood in red, both for the reduced problem, i.e., the one with latent states. Equation (5) to my understanding refers to the case without latent states. $\Lambda$ in Eq.(5) must be replaced by $\lambda\Gamma$ to compute the likelihood of the reduced model.

    **Change:**

    **Answer:**

That is correct, and precisely what is meant. The exact log-likelihood is in the formula (5), and $\Lambda = \lambda\Gamma$ should be inserted therein. We clarified the text.

28. l.209 The only parameter in the algorithmic procedure introduced above is the reduced process dimension K for the number of collective causality boxes.

    You have not used the term 'causality boxes' before. Please introduce properly.

    Also, what about 'm', the number of subdomains?

    By the way, 'm' might not be the best choice for the number of subdomains, since in Sec. 'm' was introduced as the number of categories for the output variable.

    **Change:**

    **Answer:**

    The term "causality boxes" is replaced by "latent states" everywhere. We changed denomination here. We call the number of grid points in which we consider the drafts $a_m$ and we use $m$ as number of output categories.

29. Figure 3 – Caption: Again, Eq.(5) refers to the full model without latent states. Also, it is not clear what subdomains have been chosen to generate the results presented in the Figure.

    **Change:**

    **Answer:**

    We clarified the text here, this is the same confusion as in R2SC27 above.

30. Sec. 4.1.1: I do not understand, why the content of this section is presented under 'results' and not in the data pre- and postprocessing section. I think this would nicely fit into into 3.2.2, where the classification of $w_i$ values is discussed in first place.

    The histogram presented in Fig. 4 is not really a result of this work, but can legitimately be considered as part of preprocessing.

    **Change:**

    **Answer:**

    We took another look at the structure of the entire work. The subsection 4.1 labeled 'results' belongs to the pre- and postprocessing in subsection 3.2. The

title 'results' is misleading. The histogram presented in Fig. 3 is part of the preprocessing and was redone. Moreover, the caption was rewritten.

31. l.216 In the following, the pre- and postprocessing on DBMR with respect to the categorical in- and output will be discussed.

    Why is this still 'pre-and postprocessing' if it is presented in the 'Resuts' section?

    Also, what means 'pre-and postprocessing on DBMR'?

    **Change:**

    **Answer:**

    We have deleted the sentence.

32. l.218 All-day mean data serve as basis for determining the interval for vertical draft.

    However, 'the interval' is an interval for the value of $w_i$ that is classified as 'no draft'. In the next sentence you refer to a 'subclassification'. To me it is not quite clear, what exactly is being subclassified 'by the interval'.

    If possible, please reformulate in a more precise manner.

    **Change:**

    **Answer:**

    We have rewritten the paragraph.

33. Fig.4 What exactly is shown here? In Sec. 3.1 you write that you use 12h averaged data, but the in caption it says hourly averaged data? Does the word 'mean' in 'mean vertical velocities' refer to a spatial mean over 125km? The data resolution is 6km! 'vertical velocities for day and night' contradicts 'hourly averaged values'.

    Please be more precise!

    Red vertical lines represent a tube for NO vertical draft.

    The summer months July and August in the years 1995 to 2015.

    In captions it might be acceptable to write incomplete sentences, but this 'statement' is not convenient.

    The sample size of the reanalysis data set sums up to S = 1302 (2 times 31 times 21).

    Yes, but that is not what you are showing. Why do you actually consider hourly resolved data to define a classification for the 12h averaged data?

Also, you do not give any justification for the choice of the 'no draft' – interval. Please elaborate on the criteria that you apply. Is it a certain percentile in the data?

**Change:**

Pages 10-11

**Answer:**

We have elaborated on the criteria that we apply. 12h averaged data are used based on hourly raw data for Fig. 3 and 4. We have rewritten the caption and have redone the figure. The denomination "tube" is misleading, we deleted that. The red lines are the interval for vertical draft. We explained how these were chosen and for which percentile.

34. l.225 On each box, the central mark indicates the median, and the bottom and top edges of the box indicate the 25th and 75th percentiles of CAPE concerning the affiliated n categories (n = 10 divided by quantiles).

What is meant by 'concerning the affiliated n categories'?

**Change:**

**Answer:**

We deleted "concerning the affiliated n categories" in this sentence. We added a sentence afterwards: We used n=10 categories for CAPE.

35. l.227 In Fig. 5 one sees that the input categorization is similar in terms of value for day and night. The first latent state includes 5 (for day) and 4 (at night) CAPE categories with high values. This represents high CAPE values and is therefore referred to as "High". Five (for day day) and 6 (at night) categories are affiliated to the second latent state, which is denoted with "Low".

What criterion are these assignments based on? How are the latent states defined?

I think I understand this better having read the entire manuscript. However, the fact that first a manual categorization is performed and that the DBMR performs a second categorization might be the cause of some confusion here. This could be avoided if you pointed out these two levels of categorization at an earlier stage.

**Change:**

**Answer:**

The latent states are found by the algorithm itself. The latent states can be seen as information entropic equilibria with respect to the meteorological application.

36. Figure 5. Top: Boxplot of CAPE categories by 2 latent states for daily mean (left: day and right: night); Bottom: Affiliation of CAPE categories to the latent states; Data on large scale: 500 km times 500 km subdomains for northwest of Germany, hourly averaged CAPE; On mesoscale: 125 km times 125 km subdomains for vertical velocity. Time series: 21 years for July and August, S = 1302 (Length of time series)

I understand that in the top two panels, the boxplots show the 12h averaged CAPE data (spatially averaged over the Northwest quadrant of the COSMO-REA6 data) which is assigned to the latent states 'high' and 'low' on the left and right of each of the two top panels, respectively. The left panel shows the 'day' data and the right panel the 'night' data. However, this interpretation contradicts the fact, that the distributions shown for the 'high' and 'low' latent states do in fact overlap. Please clarify, what distributions are shown here?

This is not quite clear from the caption. Please use (a), (b) ... to unambiguously mark the panels.

What role plays 'hourly averaged CAPE' in this plot?

What do the red crosses above the 75 quantile of the box plots mean? Are these extreme events? Why are there none below the 25 quantile?

What does 'On mesoscale: 125 km times 125 km subdomains for vertical velocity.' mean in the context of this plot? No vertical velocity data is shown here.

**Change:**

Pages 12-13

**Answer:**

We have rewritten the caption and paragraph.

37. l.233 The difference between the scales is small (375 km) with 500 km step size on large scale and 125 km step size on the smaller scale. The scale jump is of factor 4 on the basis of the small scale.

This is unclear. I understand that you somehow want to relate a $(125km)^2$ averaged vertical velocity data set with the $(500km)^2$ averaged CAPE data. But the $(125km)^2$ averaged vertical velocity data was not specifically introduced before? You are still at the stage, where you define the latent states for the categorical input data, why would you discuss specific choices for the output data resolution, here?

Please clarify at the beginning of the section, that a) you now move to the application of the DBMR algorithm and b) that this application requires a choice of the scale of the categorical output or in other words, a choice of the spatial average taken on the vertical velocity data. Please clarify that for the first DBMR application the spatial scale of the output is set to 125km.

Also, it would help to emphasize, that the choice of the spatial scale for the categorical output will influence the latent states identified by the DBMR.

**Change:**

**Answer:**

We added a sentence: "The choice of the spatial scale for the categorical output will influence the latent states identified by the DBMR."

38. l.245 In Fig. 6, K bivariate histograms are shown for day and night respectively. This is a specific case where apparently K=2, so please indicate.

**Change:**

**Answer:**

We have changed the sentence: "In Fig. 6, K=2 bivariate histograms are shown for day and night respectively."

39. Figure 6: Please use a discretized colorbar. Also check if other colorbars facilitate a better distinction between the many low values. Please use (a), (b) ,... to identify panels

To which quadrant of the COSMO-REA6 data do the results refer to?

**Change:**

**Answer:**

We included plots with discrete colorbar with blocks. Moreover, we have rewritten the caption.

40. l. 256 In the top left panel (Z 1 , day) of Fig. 6 probability adds up for numbers of up- or downdrafts higher than 10 to 81.

Judging from an optical assessment there should be at least as much probability weight allocated to states where both, the number of downdraft cells and the number of updraft cells are below 10. This contradicts the above statement. Please check.

**Change:**

**Answer:**

We have rewritten the sentence: "In the top left panel (Z 1 , day) of Fig. 6 probability adds up for numbers of updrafts below 10 to 81%".

41. l.257 In the top right panel, probability accumulates at small numbers of boxes with downdraft.

I cannot see any of the states with pronounced downdraft having high probability! Much of the probability mass is allocated to states with no downdraft, and little updraft.

**Change:**

**Answer:**

We have changed the sentence to: "In the top right panel, much of the probability mass is allocated to states with no downdraft, and little updraft".

42. l.262 In the bottom left panel (Z 2 , night) the probability is accumulated to 82 for the number of updrafts between 0 and 4.

That's the bottom right panel.

**Change:**

**Answer:**

We have changed the sentence to: "In the bottom right panel ($Z_2$, night) the probability is accumulated to 82% for the number of updrafts between 0 and 4.".

43. Figure 7: What is the difference between Fig.5 and Fig.7 ? If it is, that you use CAPE values averaged spatially averaged over $(125km)^2$ in fig.5 and $(15km)^2$ in Fig. 7, then this is not clear and also contradicts the statements in sec. 3:

The domain that covers Germany in Fig. 2 is divided into four 500 km times 500 km quadrants, where the spatial arithmetic mean

of each of the quadrant is considered such that we obtain one CAPE value for each quadrant.

We use the average of the 500 km times 500 km

quadrants, considering CAPE as the large scale atmospheric driver.

**Change:**

**Answer:**

We have rewritten the caption of Fig. 7.

44. l.270 KDE is a non-parametric way to estimate the probability density function of a random variable.

A KDE requires the choice of a bandwidth and is therefore not non-parametric.

**Change:**

**Answer:**

We have delted the sentence.

45. l.283 Affiliations without gaps lead to a separation of the latent states. "No gaps" means that affiliations are interrelated and not interrupted in the middle plots of Fig. A1 and Fig. A3. The affiliations have no gaps for day and night.

Of course the latent states should cover all categorical input states and should have 'no gaps'. Or does 'no gaps' mean that there is an overlap of spatially smaller scale CAPE values which are assigned to the different latent states based on the large scale CAPE average?

**Change:**

**Answer:**

"No gaps" means that there is no overlap and a clear separation of the latent states regarding the range of CAPE. This does not apply to every run with DBMR. Here we show the best ML bound estimate of 100 runs.

46. l.343 The representation of probabilities of numbers of updrafts and downdrafts conditioned on the latent states for the convective scale in Fig. 8 correspond in their distributions to the results on mesoscale in Fig. 6.

'for the convective scale' should refer to 'probabilities of numbers of updrafts and downdraft', right? In the current version of the sentence, 'for the convective scale' seemingly refers to 'the latent state'.

**Change:**

**Answer:**

We changed the sentence to: "The representation of probabilities of numbers of updrafts and downdrafts conditioned on the latent states in Fig. 8 correspond in their distributions to the results on mesoscale in Fig. 6."

47. l.355 The stochastic method is tested in a meteorological application towards a model reduction to latent states of smaller scale convective activity conditioned on large scale atmopsheric flow.

'latent states of smaller scale convective activity' suggests that the latent states are introduce for the vertical velocity, however, they have been introduced for the caterogical input, which is the large scale atmospheric flow, that is CAPE.

**Change:**

**Answer:**

We have rewritten the sentence: "In this work, a direct Bayesian model reduction of smaller scale convective activity conditioned on large scale dynamics is investigated with regard to intermediate latent states."

**2.3 Technical comments (TC)**

1. l.26 Since its introduction by Lorenz (1956), EOF analysis—known as principal component analysis (PCA) or proper orthogonal decomposition (POD)—has become an important statistical tool in atmosphere science.

- also known ... -

**Changes:**

**Answer:**

We changed the sentence to: "Since its introduction by Lorenz (1956), EOF analysis—also known as principal component analysis (PCA) or proper orthogonal decomposition (POD)—has become an important statistical tool in atmosphere science."

2. l.37 The latter approach does not require a distributional assumption but works instead with a discretized state vector.

Why 'the latter'?

**Changes:**

**Answer:**

We changed the sentence to: "The approach does not require a distributional assumption but works instead with a discretized state vector."

3. l.43 Various energetic variable are applicable on large scale.

variables is missing an s.

**Changes:**

**Answer:**

We changed the sentence: "Various energetic variables are applicable on large scale."

4. l.48 In Sect. 4 the results are discussed related to atmospheric dynamics.

... the results are discussed and related to

or the results are discussed with regard to atmospheric dynamics.

**Changes:**

Pages 2

**Answer:**

We have changed the sentence to: "In Sect. 4 the results are discussed with regard to atmospheric dynamics."

5. l.114 Moreover, the method yields $\Gamma_{kj}$, i.e., the original input categories are assigned to the reduced system's (latent) categories in a deterministic fashion (no "fuzzyness" in the affiliations).

Why is the $\Gamma_{kj}$ marked with a star?

**Changes:**

None

**Answer:**

The star marks the optimal solution/affiliation; see Eq. (9).

6. l.120 This manifests in the variance of the estimated parameter lamda, which shows a K/n-times smaller uncertainty than lamda...

   I am not sure the use of 'manifest' is correct here.

   **Changes:**

   **Answer:**

   We have reformulated the sentence.

7. l.155 The sample size of the reanalysis data set used in Sect. 2 sums up to S = 1302 (2 times 31 times 21)

   Sect. 2? Wrong reference?

   **Changes:**

   Pages 7

   **Answer:**

   We have rewritten the sentence: "The sample size of the reanalysis data set used in Sect. 2.2 sums up to S = 1302 (2 times 31 times 21)". You find S on page 4, line 93.

8. l.168 According to the meteorological data in Sect. 3.1...

   According to the meteorological data described in Sect. 3.1...

   **Changes:**

   **Answer:**

   We have written "According to the meteorological data described in...".

9. l.169 CAPE plays the role of an input variable X in Sect. 2

   CAPE plays the role of an input variable X as defined in Sect. 2

   **Changes:**

   **Answer:**

   We have reformulated the sentence.

10. l.175 With this type of classification, extreme weather events tend to be in a separate category.

tend to be in separate categories.

**Change:** Page 9

**Answer:**

We have deleted the sentence. We just present one type of classification via quantiles.

11. l.175 These are not Gaussian distributed.

Grammatically the reference of 'these' is not clear. Could refer to 'categories' from the previous sentence. I assume you meant to refer to 'extreme weather events'. Please clarify and add 'in terms of CAPE'.

**Changes:**

**Answer:**

We have deleted the sentence.

12. l.193 Let $Y_i(t)$ be the discretized vertical velocities at time t with 1 ¡ i ¡ m numbering the grid boxes on the corresponding scale, see Tab. ??.

Probably, it is better to not use m as an upper bound for i, since m was already use for the number of observations of Y in time. Here, m refers to the spatial number of observations that are being made simultaneously, if I understand correctly.

Please correct the reference.

**Changes:**

**Answer:**

We have changed the notation for the spatial number of observations and corrected the reference.

13. l.206 for every fixed number K of latent state.

for every fixed number K of latent states.

**Changes:**

Pages 10

**Answer:**

We changed "state" to "states".

14. l.206 For the respective latent state...

For each K.

'The latent state' would refer to an individual state.

**Changes:**

Pages x, lines xxx

**Answer:**

We have written "For each K".

15. l.220 In Fig. 4, the histogram of mean vertical velocities for a resolution of 125km with the interval is shown.

In Fig. 4, the histogram of mean vertical velocities for a resolution of 125km is shown together with the interval that defines the 'no draft' category.

**Changes:**

**Answer:**

We have changed the sentence.

16. l.229 Five (for day day) and 6 (at night) categories

double day

**Changes:**

**Answer:**

We have deleted one "day".

17. l.244 In order to visualize the probabilities of the small scale conditioned on the latent states. . .

In order to visualize the probabilities of the small scale variable conditioned on the latent states of the large scale variable. . .

**Changes:**

Pages 12

**Answer:**

We changed the sentence.

18. l.244 the entries of the lamda

Why does $\lambda$ carry a hat?

**Changes:**

**Answer:**

No hat.